# On the importance of moisture conveyor belts from the tropical East Pacific for wetter conditions in the Atacama Desert during the Mid-Pliocene

Mark Reyers[1], Stephanie Fiedler[1,2], Patrick Ludwig[3], Christoph Böhm[1], Volker Wennrich[4], Yaping Shao[1]

[1]Institute for Geophysics and Meteorology, University of Cologne, Cologne, Germany

[2]now: GEOMAR Helmholtz Centre for Ocean Research Kiel and Faculty of Mathematics and Natural Sciences, University of Kiel, Kiel, Germany

[3]Institute of Meteorology and Climate Research, Karlsruhe Institute of Technology, Karlsruhe, Germany

[4]Institute for Geology and Mineralogy, University of Cologne, Cologne, Germany

*Correspondence to*: Stephanie Fiedler (sfiedler@geomar.de)

**Abstract.** Geomorphic and sedimentologic data indicate that the climate of today's hyper-arid Atacama Desert (Northern Chile) was more humid during the Mid-to Late Pliocene. The processes, however, leading to increased rainfall in this period are largely unknown. To uncover these processes we use both global and regional kilometre-scale model experiments for the mid-Pliocene (3.2 Ma BP). We found that the PMIP4-CMIP6 model (CESM2) and the regional model (WRF) used in our study simulate more rainfall in the Atacama Desert for the mid-Pliocene in accordance to proxy data, mainly due to stronger extreme rainfall events in winter. Case studies reveal that these extreme winter rainfall events during the mid-Pliocene are associated with strong moisture conveyer belts (MCBs) originating in the tropical East Pacific. For present-day conditions, in contrast, our simulations suggest that the moisture fluxes rather arise from the subtropical Pacific region and are much weaker. A clustering approach reveals systematic differences between the moisture fluxes in the present-day and mid-Pliocene climates, both in strength and origins. The two mid-Pliocene clusters representing tropical MCBs and occurring less than one day per year on average produce more rainfall in the hyper-arid core of the Atacama Desert south of 20°S than what is simulated for the entire present-day period. We thus conclude that MCBs are mainly responsible for enhanced rainfall during the mid-Pliocene. There is also a strong SST increase in the tropical East Pacific and along the Atacama coast for the mid-Pliocene. It suggests that a warmer ocean in combination with stronger mid-tropospheric troughs is beneficial for the development of MCBs leading to more extreme rainfall in a +3K warmer world like in the mid-Pliocene.

## 1 Introduction

The Atacama Desert in Northern Chile (19°S – 26°S) is considered to be one of the driest deserts on Earth, with a mean annual rainfall of only a few mm in the hyper-arid core and along the Coastal Cordillera (e.g., Houston, 2006; Reyers, et al. 2021). Due to its special geographic position, the aridity in the Atacama is long-lasting, with its onset likely dating back to the Oligocene (ca. 33.9- 23 Ma ago), thus making it also one of the oldest deserts (e.g. Dunai et al., 2005; Evenstar et al., 2017). Paleoclimate records and geomorphic studies from various sites of the Atacama Desert, however, indicate variations in the overall aridity over time, which was repeatedly punctuated by more humid phases on millennial and orbital time scales (Ritter et al., 2019; Diederich et al., 2020). Based on geomorphic and sedimentologic data, Amundson et al. (2012) and Hartley and Chong (2002) postulated that nowadays dominantly hyper-arid conditions commenced during the late Pliocene (ca. 3-2 Ma ago). This shift to overall hyper-aridity postdates a period of more humid or even semi-arid conditions in the Atacama Desert during the Mid-to Late Pliocene, that is illustrated in the widespread occurrence of lake, salar, and fluvial deposits in many parts of the desert (e.g., Gaupp et al., 1999; Kirk-Lawlor et al., 2013; Jordan et al., 2014; Evenstar et al., 2016; Ritter et al., 2018; Vásquez et al 2018). While such proxy records provide a reliable picture of the timing and magnitude of past pluvial phases, the mechanisms controlling these climate shifts and in particular the involved atmospheric processes are hard to derive

from them, and are thus not well understood. Intervals of increased rainfall in the Southern Atacama Desert are mostly attributed to a northward displacement of mid-latitudinal westerlies and accompanied extra-tropical winter cyclones (Vuille and Ammann, 1997, Stuut and Lamy, 2017; Bartz et al., 2019), which suggest a southwestern moisture source. Past rainfall variations in the Northern Atacama and the Andes have been linked to latitudinal shifts of the extra-tropical westerlies in the Southern Hemisphere (Amidon et al., 2017). For the warmer mid-Pliocene climate, the multi-model mean of the PlioMIP2 models for instance indicate that the Hadley Cell was shifted northward and the Walker Circulation shifted westward (e.g. Han et al., 2021). Also, cut-off lows as seen in March 2015 have been proposed as possible mechanism for wetter conditions in the past (Jordan et al., 2019). Wetter conditions during the warm Pliocene are furthermore attributed to a mean state more similar to today's El Nino conditions, and according to this the onset of the hyper-aridity concurrently occurred with the change to the present-day ENSO variability (Amundson et al., 2012). Hartley and Chong (2002) concluded that the shift from semi-arid to hyper-arid conditions in the Atacama was controlled by global cooling rather than by local factors, like the cold Humboldt Current.

Another perspective on potential drivers for wetter paleoclimate episodes can be inferred from the mechanisms involved in the March 2015 extreme rainfall event that caused severe damage in the Atacama Desert. The event was associated with a mid-tropospheric cutoff low and anomalously warm tropical sea surface temperatures (SST) along the Atacama coast, paired with transport of large amounts of water vapor from the tropical East Pacific to the Atacama Desert at the foreside of the low-pressure system (Bozkurt et al., 2016). Based on the characteristic isotopic composition of rain water (e.g., Herrera and Custodio, 2014) from the March 2015 event, Jordan et al. (2019) proposed that the processes involved in this event might also play an important role in increased paleoclimate rainfall in the Atacama Desert.

Böhm et al. (2021, hereafter B2021) systematically investigated moisture conveyer belts (MCBs) in the Atacama for the present-day climate that also occurred during the March 2015 event. MCBs are elongated bands of strong poleward water vapor fluxes in the free troposphere (e.g., Knippertz and Martin, 2007), which often originate in tropical latitudes and occur in connection with troughs or cutoff lows. Based on different datasets for the recent past, B2021 indicate that moisture transport associated with MCBs is decoupled from the maritime boundary layer. Despite their rareness of about four regional MCBs per year which make landfall in the Atacama Desert, MCBs produce 40-80% of the total rainfall in the hyper-arid core and along the Coastal Cordillera for modern climate conditions (B2021).

The aim of the present study is to assess to what extent MCBs act as drivers of rainfall activity in the Atacama Desert on geological time scales, specifically during the relatively wet and warm mid-Pliocene period. If MCBs played an important role in that period, this would imply that in addition to the previously suggested regions southwest or east of the Atacama Desert (Stuut and Lamy, 2017; Bartz et al., 2019; Amidon et al., 2017) also the tropical Southeast Pacific northwest of the desert could be a potential moisture source for increased humidity in the mid-Pliocene, like assessments of the regional rainfall under present-day climate suggest (Bozkurt et al., 2016, Jordan et al., 2019; Böhm et al., 2021). However, the past and present

constellations of the global atmospheric and oceanic circulations are substantially different. The Atacama Desert is characterized by a complex topography, including a steep coastal cliff and the western slopes of the Andes, shown in Fig. 1. As a consequence, strong rainfall gradients occur, in particular in west-to-east direction (Houston, 2006; Reyers et al., 2021). Due to the rather coarse horizontal resolution of global climate models, the kilometre-scale changes in the orography and rainfall patterns cannot be adequately simulated by design (e.g., Ludwig et al., 2019; Fiedler et al., 2020). We therefore perform kilometre-scale simulations for a limited-area domain over the Atacama Desert for the mid-Pliocene at around.3.2 Ma. Section 2 gives an overview of the datasets and methods used in our study. Section 3 consists of two subsections, where we present large-scale and regional paleo-climate changes as obtained by global and downscaled PMIP4-CMIP6 models (section 3.1), and potential key driver for the detected rainfall changes (section 3.2). A summary and discussion section closes this paper (section 4).

## 2 Data and Methods

### 2.1 Model simulations

We use global climate model (GCM) output from the fourth phase of the Paleoclimate Modeling Intercomparison Project, which is endorsed by the sixth phase of the Coupled Model Intercomparison Project (PMIP4-CMIP6; Kageyama et al., 2018). Here, we examine the output of a mid-Pliocene Warm Period experiment (3.2 Ma before present), as part of the Pliocene Model Intercomparison Project (PlioMIP2; Haywood et al., 2016) Phase 2. In the PlioMIP2 simulations, the $CO_2$ concentration is set to 400 ppm, while other greenhouse-gas concentrations and the orbital parameters are as for the pre-industrial period (1850) to be consistent with the setup of PlioMIP2 experiments. Paleoenvironmental reconstructions from the Pliocene Research, Interpretation and Synoptic Mapping project Phase 4 (PRISM4) are used as paleogeography boundary conditions, e.g., topography, bathymetry, ice sheets and vegetation cover. A significant feature of these reconstructions is the strong reduction of the Greenland and Antarctic ice sheets compared to today. Dowsett et al. (2016) describes PRISM4 in detail.

From PlioMIP2 we select simulations from the Community Earth System Model Version 2 (CESM2) of the National Center for Atmospheric Research (NCAR) (Danabasoglu et al., 2020) for our study. This choice has been made based on the availability of boundary data and the validation results of GCMs participating in PlioMIP2, assessed previously (e.g., Feng et al., 2020, Pontes et al., 2020, Han et al., 2021). In addition to CESM2, two other PlioMIP2 / CMIP6 models are analysed, namely IPSL-CM6A-LR (Boucher et al., 2018) from the Institut Pierre Simon Laplace and GISS-E2.1-G (Bauer et al., 2020) from the NASA. Only for these models, the historical and the mid-Pliocene experiment are available. Our PlioMIP2 inter-comparison for the Atacama revealed that

(i) CESM2 reproduces the increased rainfall for the Atacama region compared to present-day in accordance with proxy data (Sections 3.1), and

(ii) CESM2 shows an expected SST increase for the mid-Pliocene, which qualitatively agrees with reconstructions (Section 3.1).

This makes the CESM2 output the most useful amongst the three GCMs from PlioMIP2 to perform the high-resolution simulations for the Atacama Desert.

For the atmosphere CESM2 has a horizontal resolution of 1.25° x 0.9° in longitudinal and latitudinal direction, respectively. The horizontal resolution of the ocean is constant in the zonal direction (1.125°) and varies from 0.27° to 0.64° in the meridional direction. We select a 30-year period in the mid-Pliocene experiment of CESM2 that includes a strong ENSO and rainfall variability (model years 231-260) to represent internal variability. To quantify changes relative to present day, we further use a 30-year time slice of the CMIP6 historical experiment (years 1985-2014) of CESM2.

The output of CESM2 is used as initialisation and boundary data for simulations with the Weather Research and Forecasting Model (WRF; http://www.wrf-model.org; Skamarock et al., 2008) Version 3.9. The lateral boundary fields are 6-hourly updated. We yield a horizontal resolution of 10km x 10km via a double one-way nesting: $1^{st}$ nesting from 1.25° x 0.9° to 50 x 50km and $2^{nd}$ nesting from 50 x 50km to 10 x 10km. The outer and inner model domains for our WRF simulations are depicted in Fig. 1. For our analysis we use the output of both nesting steps (i.e. WRF simulations with 50 and 10 km, respectively).

Reyers et al. (2021) demonstrate that the WRF model reproduces present-day rainfall in the Atacama Desert when driven with reanalysis data, despite the complexity of the terrain. We therefore use the same model setup in terms of the physical parameterisations for this study on the mid-Pliocene climate. The WRF simulation uses prescribed time-varying SSTs from CESM2 that are updated every 24 hours. The same 30-year mid-Pliocene (mP) and historical (hist) periods as used for the analysis of the global CESM2 model are simulated with WRF (hereafter $WRF_{mP}$ and $WRF_{hist}$). For the $WRF_{mP}$ simulation the

greenhouse-gas concentrations and the orbital parameters are prescribed as in the global PlioMIP2 simulations (see above). Paleo-records reveal that the land cover strongly changed during the Pliocene on global scales, but a reconstruction for South America is difficult due to the sparse data coverage (Dowsett et al., 2016). As a consequence, reconstructed land cover for the Atacama Desert is implausible. To avoid uncertainties arising from artefacts in reconstructed land-cover changes and for a better comparability of our WRF simulations, we therefore prescribe the same present-day vegetation mask in all WRF

simulations. For an additional evaluation of the WRF simulations for the Atacama, rainfall of $WRF_{hist}$ is compared to the present-day validation simulation with WRF from Reyers et al. (2021), which was driven by ERA-Interim reanalysis data ($WRF_{era}$).

## 2.2 Analysis strategy

For the assessment of the dynamical processes involved in rainfall in the Atacama Desert we first quantify atmospheric moisture fluxes from the WRF simulations. This is done by computing the integrated water vapor flux (IWVF), which consists of a zonal ($IWVF_u$) and a meridional ($IWVF_v$) component:

$$IWVF_u = \int_{z1}^{z2} uq\,dz, \tag{1}$$

$$IWVF_v = \int_{z1}^{z2} vq\,dz, \tag{2}$$

where $q$ is the specific humidity, $z$ is the height, and $u$ and $v$ are the zonal and the meridional wind components, respectively. B2021 demonstrate that MCBs are associated with elevated moisture transport decoupled from the maritime boundary layer. To take this into account we compute IWVF for low levels between 0 and 2000m above ground level (agl.), and for upper levels above 2000m agl., separately.

For the classification of MCBs in our study area, we apply machine learning techniques to the IWVF over the offshore region of western central South America. Due to the rareness of MCBs in the Atacama Desert for the present-day climate (B2021), we choose a method that can detect outliers. This is a combination of self-organizing maps (SOMs, Kohonen, 2001) and K-means clustering (e.g., Hartigan and Wong, 1979). This combined method has already been applied successfully in climate research (e.g., Ohba et al., 2016). SOMs is a machine learning technique often used for pattern recognition. In this technique, high-dimensional data is projected onto a visually interpretable two-dimensional grid by mapping similar input vectors close to each other. As input vectors we use 12-hourly IWVF patterns as diagnosed from our WRF simulations. IWVF values below a certain threshold are set to zero before the SOM clustering. This threshold-based methodology is inspired by approaches which are used to identify low-level atmospheric rivers (e.g., Rutz et al., 2014), and ensures that aside from the strength of the MCBs also information about their shape is provided for the clustering. For $WRF_{hist}$ an IWVF threshold of 250 kg·m$^{-1}$·s$^{-1}$ is chosen. Due to the fact that during the warmer mid-Pliocene the atmosphere is generally moister, the threshold for $WRF_{mP}$ is set to a higher value of 350 kg·m$^{-1}$·s$^{-1}$. Sensitivity tests indicate that slight differences in the thresholds yield similar clustering results from the clustering, although the choice of these threshold values is somewhat arbitrary. The input vectors are projected on a 7 x 7 grid SOM, thus resulting in 49 IWVF classes. To further reduce this data, the 49 IWVF classes are clustered via K-means. K-means clustering is an iterative algorithm which obtains an optimal partition of the input data (here, the 49 IWVF classes from the SOM) into $k$ clusters. The algorithm computes so-called cluster centroids, and an optimal solution is found when the squared distances between the cluster members and their respective cluster centroids are minimized. The resulting cluster centroids represent the final clusters. For a better comparability, we aim for the same number of final clusters for $WRF_{hist}$ and $WRF_{mP}$. Nine final clusters are optimal, since the squared distances between the centroids converge towards a minimum value for more than nine cluster for both, $WRF_{hist}$ and $WRF_{mP}$. Note that the order of clusters is arbitrary.

For the identification of mid-tropospheric troughs off the Atacama coast we adopt the automated identification of Knippertz (2004). A grid point $P$ is defined as a trough point, where the west-to-east gradient in the 500-hPa geopotential height exceeds a threshold of 25 gpm per 10°. Here, we apply the detection algorithm to the CESM2 data instead of the regionally limited WRF output since a larger spatial coverage is needed for identifying synoptic-scale troughs. The 500-hPa geopotential height is interpolated to a regular 2°x2° horizontal grid first. The interpolation is done for consistency with the coarse reanalysis data used in the original development of the detection algorithm.

## 3 Results

### 3.1 Mid-Pliocene against present-day climate

The Pliocene is considered to be the most recent period in Earth history with globally sustained higher temperatures by several degree Celsius, and thus a potential analogue for the conditions projected by CMIP6 scenarios for the future. During the stable mid-Pliocene period between 3.264 and 3.025 Ma the global annual mean temperature is, based on climate models, assumed to be up to more than 3°C warmer than in present-day (Haywood et al., 2013; Haywood and Valdes, 2004). The model results are supported by proxy data indicating a global SST anomaly for the mid-Pliocene vs. pre-industrial of 2.3°C and 3.2–3.4°C based on foraminifera Mg/Ca and alkenones or alkenones only, respectively (McClymont et al., 2020). Specifically in the upwelling regions at the Peruvian margin, Deckens et al. (2007) reconstructed a Pliocene-modern SST change by 2.9°C. Fig. 2a shows the mean annual near-surface air temperature changes for the mid-Pliocene against present-day (historical experiment) from CESM2. A strong polar amplification is visible over Antarctica, with mid-Pliocene temperatures being partly more than 10°C warmer than for present-day conditions. A particular strong warming is also revealed in the tropical East Pacific and along the western coast of South America, which is associated with changes of the SST in the tropical East Pacific and in the upwelling zone off the Chilean coast. Here, the SST is simulated to be up to 5°C warmer during the mid-Pliocene (Fig. 2b). This SST difference pattern qualitatively agrees with the PRISM reconstructions for this area (Dowsett et al. 2013; see their Figure 5). A strong precipitation increase is found for the Altiplano region, where mean annual rainfall in the mid-Pliocene is up to 400 mm higher than in the historical experiment (Fig. 2c). More rainfall is also simulated for the Atacama Desert. These results for more rainfall are broadly consistent with proxy records for the wetter conditions in the mid-Pliocene compared to present-day conditions in the region, listed in Table 1.

In contrast, IPSL-CM6A-LR and GISS-E2.1-G simulate dryer conditions in the Atacama Desert for the mid-Pliocene compared to present-day (see Supplementary Fig. S1c and S2c). Both models also exhibit a strong polar amplification like CESM2, but the warming in the Southern Hemisphere mid-latitudes and tropics is much weaker when compared to CESM2 (Supplementary Fig. S1a and S2a). In particular the atmospheric and surface temperature increase in the tropical East Pacific and off the Atacama coast is less pronounced in these two models (Supplementary Fig. S1a,b and S2a,b) compared to CESM2. Hence, amongst the three available PlioMIP2 models the CESM2 model has the best agreement with proxy data regarding both, SST, and rainfall in the study area.

We evaluated the rainfall from WRF$_{hist}$ against a WRF simulation that downscales the ERA5 reanalysis for the same domain and spatial resolution (WRF$_{era}$). There are quantitative differences in rainfall, but the aridity is overall satisfyingly reproduced by the WRF simulation that used data from the historical simulation of CESM2 at the lateral boundaries (WRF$_{hist}$ ). Specifically, the spatial patterns and the seasonal cycle of rainfall are qualitatively captured by WRF$_{hist}$ (compare Fig. 3f-j with Fig. 3a-e). Both WRF$_{era}$ and WRF$_{hist}$, show the present-day hyper-aridity in the Atacama Desert, with mean annual rainfall of less than 2mm·yr$^{-1}$ north of 22°S. Rainfall slightly increases towards the south, but a particular strong west-to-east gradient

related to the topographic characteristics of the region is visible in the north-eastern part of the model domain. The rainfall at the western slopes of the Andes is strongest in summer (DJF), whereas strongest rainfall in the hyper-arid core of the Atacama is simulated for winter (JJA). Annual and seasonal rainfall amounts tend to be regionally overestimated by $WRF_{hist}$ against $WRF_{era}$, but the hyper-aridity with only a few mm of rainfall per year is well simulated (Fig. 3f-j). We therefore conclude that the WRF simulations using CESM2 as boundary conditions are suitable for our research interest.

For the mid-Pliocene ($WRF_{mP}$, Fig. 3k), our simulations yield more humid conditions, with nearly a doubling in the annual rainfall for most parts of the desert south of 20°S compared to $WRF_{hist}$ (compare Fig. 3k and 3f). This mid-Pliocene rainfall increase in the hyper-arid core of the Atacama Desert is mainly due to more winter rainfall (Jun-Aug, Fig. 3n). We therefore focus on this season in the remainder of our study and first test the two following hypotheses:

(i) the intensity of rain days is similar in both periods, but there are more rain days in $WRF_{mP}$ compared to $WRF_{hist}$, and

(ii) individual rain days are more intensive in $WRF_{mP}$ than in $WRF_{hist}$.

The seasonal cycle of rainfall shifts from winter-dominated in the south to summer-dominated in the north of the Atacama Desert (Houston, 2006; Reyers et al., 2020). In order to account for these differences, we split the hyper-arid core into a northern and southern region (see black boxes in Fig. 3n) which has also been done similarly in previous studies (e.g. Böhm et al., 2020). To quantify the cause of the winter rainfall increase (see Fig. 3n), we compute the spatial means of daily rainfall over these two regions. Then we rank the spatially averaged daily rainfall amounts according to their magnitude and display the results in percentile-percentile plots for the mid-Pliocene against the present-day simulation (Fig. 4). For both, the northern and the southern boxes, there is no increase in the number of rainfall events in $WRF_{mP}$ when compared to $WRF_{hist}$, and appreciable rainfall amounts are rare in both simulations. Instead, rain days at the upper end of the distribution are stronger during the mid-Pliocene. For example, for the northern (southern) box daily rainfall is appr. 12 mm (55 mm) per grid point on average for the strongest event in $WRF_{mP}$, but only 5 mm (21 mm) for the strongest event in $WRF_{hist}$. Hence, our simulations suggest that the more humid conditions during the mid-Pliocene is due to rain days with more intense rainfall events by a factor of about two, not due to more rain days.

## 3.2 Potential drivers for stronger rainfall events in the mid-Pliocene

We assess physical processes, potentially driving the stronger winter rainfall events in the mid-Pliocene. To that end, we conduct several case studies of days with strong winter rainfall events in the northern and southern Atacama (see black boxes in Fig. 3n). Fig. 5 shows the rainfall patterns of exemplary top winter rainfall events as simulated by $WRF_{mP}$. It turned out that five out of the top ten rain days in the northern box are also among the top days in the southern box. As an example, two of these events are displayed in Fig. 5a and 5b. The events are associated with elongated bands of strong upper-level IWVF (above 2000m agl.), which have the typical properties of MCBs, as identified in B2021 for present-day conditions. Further, the MCBs originate in the tropical East Pacific region and transport moisture along the coastal offshore region towards the Atacama Desert. Strong rainfall events which are restricted to northern Atacama are in some cases associated with relatively

weaker MCBs, but with a similar spatial pattern with a north-west to south-east moisture flux (Fig. 5c and 5d). A rather zonal oriented moisture flux is found for one top rainfall event which is restricted to southern Atacama (Fig. 5e). The majority of the southern events, however, also coincide with strong north-westerly MCBs with origin in the tropical East Pacific, e.g., shown in Fig. 5f. These results indicate that for both, the southern and northern Atacama Desert, MCBs and thus the same physical processes are responsible for the wetter conditions in the mid-Pliocene. In contrast, upper-level IWVF during or prior the top ten events in $WRF_{hist}$ are weaker (Fig. 6). Additionally, the IWVF patterns associated with the $WRF_{hist}$ events more often show a rather zonal orientation, and thus originate from the subtropical Pacific (Fig. 6b,d,e,f).

Interestingly, low-level IWVF (below 2000m agl.) is negligible during all northern and southern top rainfall events in both, $WRF_{mP}$ and $WRF_{hist}$ (Supplementary Fig. S3 and S4), consistent with present-day cases (B2021). Upper-level moisture transport via MCBs into the study area are therefore largely decoupled from the maritime boundary layer in both present-day and mid-Pliocene conditions.

The MCBs identified in the case studies for $WRF_{mP}$ often have a Northwest to Southeast orientation (Fig. 5a,b,d,f), and are therefore associated with a southward moisture transport from the tropics. Fig. 7 shows the probability density function of 12-hourly southward upper-level IWV mass fluxes integrated between 80°W and 70°W along the 20°S transect (see yellow line in Fig. 5a) in $WRF_{mP}$ and $WRF_{hist}$. The extremely strong southward mass fluxes are more frequent in $WRF_{mP}$ than in $WRF_{hist}$ (inset in Fig. 7). For example, moisture mass fluxes of appr. $400 \cdot 10^6 \cdot kg \cdot s^{-1}$ occur more than 15 times in 30 Southern Hemisphere winters of $WRF_{mP}$ but are rare in $WRF_{hist}$. Additionally, fluxes of more than $600 \cdot 10^6 \cdot kg \cdot s^{-1}$ lacking in $WRF_{hist}$ regularly occur in $WRF_{mP}$. These findings indicate that there are cases in $WRF_{mP}$ when the large moisture reservoir over the tropical East Pacific is tapped via MCBs, advecting moist air to the Atacama Desert.

The case studies (Fig. 5 and 6) illustrate the key role of MCBs for increased rainfall in the mid-Pliocene. The question arises to what extent the characteristics of the MCBs during the mid-Pliocene were systematically different from those under present-day conditions. We perform an objective clustering of upper-level wintertime IWVF (section 2.2) to quantify potential differences. The nine final IWVF clusters for $WRF_{mP}$ are displayed in Fig. 8. The cluster with the highest frequency of occurrence is cluster 8, which represents cases without MCB and occurs at 79 days per winter on average. Instead, this cluster includes days with easterly IWVF, which in some cases is associated with deep convection over the Altiplano Plateau (Reyers et al., 2021). Consequently, a high rainfall amount in the north-eastern part of the study area is found for this cluster. Cluster 1, 2, 5, 7, and 9 include more zonally oriented IWVF with landfall in central or southern Chile with origins further towards the South of the Pacific, e.g., when compared to cluster 4. Also cluster 3 is associated with moisture advection from regions South of 20°S but includes also some transport from further North with a more tilted axis towards meridional directions compared to cluster 1, 2, 5, 7, and 9. These clusters with dominant moisture advection from regions South of 20°S play a minor role for winter rainfall in the Atacama Desert (see red-blue shading in Fig. 8). In contrast, the IWVF clusters 4 and 6, which occur approximately every second winter in $WRF_{mP}$, produce the largest rain amount during winter in the hyper-arid core of the Atacama Desert. These IWVF clusters represent strong MCBs originating in the tropical East Pacific region, which make landfall at the coastal region of the Atacama Desert. In $WRF_{hist}$, the moisture fluxes in the IWVF cluster are generally weaker

than in WRF$_{mP}$ (Fig. 9). As for WRF$_{mP}$, the more zonally oriented  IWVF clusters with landfall south of 25°S are associated with only little rainfall (cluster 1, 4, 6, 7, 8 in Fig. 9). The most crucial IWVF clusters for winter rainfall in WRF$_{hist}$ are cluster 9 and in particular cluster 3, which represent north-westerly MCBs. This is in agreement with the findings of B2021, suggesting that the moistest MCBs in the recent climate have their origin in the North-west (see Fig. 7 in B2021). However, compared to the MCBs of WRF$_{mP}$ (Fig. 8), the MCBs in WRF$_{hist}$ (in Fig. 9) have much smaller IWVF and  can have different transport characteristics, e.g.,  indicated by wind maxima that are shifted towards the South and East. Take for instance, the smaller IWV and different wind pattern in cluster 4 (6) for WRF$_{mP}$ in Fig. 8 against cluster 9 (3) for WRF$_{hist}$ in Fig. 9. The strengthening of the north-westerly MCBs in WRF$_{mP}$ might suggest that the wind speeds associated with these MCBs are higher in the mid-Pliocene leading to rainfall on land, but this is not visible in the output. We computed the composite of mean wind speeds for each of the MCB clusters associated with rain in the desert. The resulting composite of wind speeds in the clusters are marked for a typical moisture transport height of 4000m asl. in Fig. 8 and 9. Interestingly, the mean wind speeds in the two clusters in WRF$_{mP}$ have a similar magnitude like in WRF$_{hist}$. However, the region with peak winds is shifted to the north-west in WRF$_{mP}$. We therefore conclude that the MCBs during the mid-Pliocene that lead to rain in the Atacama are not only stronger, but in some cases also have origins and characteristics that are different from those that occur under present-day conditions.

Despite their rare occurrence, MCB cluster 4 and 6 of WRF$_{mP}$ (Fig. 8) are associated with a relatively large amount of rainfall. In most parts of the Atacama Desert more than 60% of the total winter rainfall in WRF$_{mP}$ is associated with only these two MCB clusters, and for some regions north of 22°S the fraction is even close to 100% (Fig. 10a). Further, in the hyper-arid core south of 20°S the mid-Pliocene winter rainfall produced by these two clusters is more than 60% of the total annual rainfall (Fig. 10b), and it even clearly exceeds the total annual rainfall in WRF$_{hist}$ (Fig. 10c). This indicates that MCBs originating in the tropical East Pacific are indeed a key driver for the more humid conditions in the Atacama Desert during the mid-Pliocene. The strength of MCBs is determined by both the winds and the atmospheric moisture content. The former is controlled by pressure gradients. We see that MCBs in the mid-Pliocene appear at the foreside of mid-tropospheric troughs over the subtropical Southeast Pacific. This is also the case for the regional MCBs for present-day conditions (B2021). Interestingly, the frequency of occurrence of troughs in the region is smaller during the mid-Pliocene compared to present day (Fig. 11a). Thermodynamic and dynamical changes in PlioMIP2 models lead to wetter tropics, particularly in the eastern tropical Pacific (Han et al., 2021). Furthermore, the subtropical anticyclones over the southern Pacific intensified and expanded further to the West in PlioMIP2 (Pontes et al., 2020). The change in the synoptic-scale circulation is a potential reason for  the smaller frequency of troughs reaching the subtropical Pacific offshore the Atacama (Fig. 11a). Fewer troughs in the warmer past is consistent with findings for future global warming, e.g., Priestley and Catto (2022) found that the Southern Hemisphere extratropical storm track density decreases under future warming based on CMIP6. However, the troughs that occur extraordinarily close to the equator (at or north of 20°S) are stronger in the mid-Pliocene, measured by the mean pressure gradients (Fig. 11b), which implies an intensification and northward shift of the flow at their northern flank. This finding is consistent with wind composites we show for the north-westerly MCB clusters (Fig. 8 and 9).

The atmospheric moisture content is influenced by the SST. In a sensitivity study, Bozkurt et al. (2016) show that a hypothetical reduction of the SST in the eastern tropical Pacific during the March 2015 rainfall event, which occurred during anomalously high SST, significantly decreases the precipitable water along the Peruvian and Chilean coast and the rainfall amount in the Atacama Desert. We therefore assess the role of the SST for the development of MCBs originating over the tropical East Pacific. To this end, we computed SST composites for all days within the MCB clusters 4 and 6 of $WRF_{mP}$. For those days, the SST composite is up to 1.2°C warmer than the median of the winter SST for the entire mid-Pliocene period (Fig. 12a). Over the equatorial and eastern tropical Pacific, the SST composite even exceeds the 70th winter SST percentile (Fig. 12b). As shown in Fig. 2b, this corresponds to the region with the strongest SST increase in the mid-Pliocene. Hence, this mid-Pliocene SST increase in combination with the stronger upper-level troughs (Fig. 11b) is a plausible scenario to explain why exceptionally strong MCBs, as identified in cluster 4 and 6 in $WRF_{mP}$, occur in the mid-Pliocene but are not seen during present-day conditions ($WRF_{hisr}$). Our results support that higher SSTs lead to stronger rainfall in the Atacama, broadly consistent with the March 2015 case studied by Bozkurt et al. (2016).

These results indicate that stronger troughs and warmer SSTs during the mid-Pliocene favour the formation of strong MCBs. It is not clear, however, how MCBs in the Atacama region develop in detail. For the Southwestern United States, Knippertz and Martin (2007) attributed the formation of an MCB to a baroclinic zone and a strong subtropical jet at the foreside of a cutoff, as well as to strong convection in the ITCZ. Although plausible, this explanation may not necessarily be applicable for MCBs in the Southeast Pacific. For the Atacama under present-day conditions, B2021 traced the air in the strongest MCBs back to the Amazon Basin. Their backward trajectories hardly indicate any moisture enrichment when passing the Pacific (see Fig. 4d of B2021), and consequently they conclude that the associated moisture most likely originates from the continental Amazon Basin. Interestingly, the isotopic analysis of the collected rain water in coastal regions from the March 2015 event indicates a tropical Pacific origin for the water (Jordan et al., 2019). While B2021 did not specifically address this particular event, they show tropical and subtropical Pacific origins for cases that are weaker in terms of humidity along the identified trajectories. The uncertainty to identify the most representative target location and time for the trajectory calculation may cause stronger events to appear weaker.

## 4 Conclusion and Outlook

We analysed global PMIP4-CMIP6 model simulations and performed regional kilometre-scale experiments to assess drivers for the increased rainfall amounts in the Atacama Desert during the warm mid-Pliocene as reported from geological evidences. Our model results suggest that the enhanced rainfall during the mid-Pliocene can be primarily explained by increased winter rainfall. Focussing on this season, we find evidence that main drivers for the increased rainfall are moisture conveyor belts (MCBs) advecting moisture from the tropical East Pacific to the Atacama Desert. This conclusion is based on the following findings:

(i) More humid conditions in the mid-Pliocene experiment are caused by stronger rainfall events that can be twice as strong compared to the present-day. The frequency of occurrence of rainfall events remains nearly unchanged.

(ii) The top mid-Pliocene rainfall events are often associated with strong elongated upper-level moisture fluxes. These share the typical characteristics of MCBs with origin in the tropical East Pacific advecting moist air to the Atacama Desert.

(iii) A clustering reveals that MCBs in the mid-Pliocene are stronger and show distinct differences in the exact origin and pathways compared to those for present-day. The two mid-Pliocene clusters for MCBs suggest that the tropical East Pacific is the moisture source, in contrast to present-day simulations (historical experiment) when these north-westerly MCBs are much weaker and their origin is further to the Southwest.

(iv) The rainfall associated with the north-westerly MCBs can be up to 100% of the regional mid-Pliocene winter rainfall, and more than 60% of the annual rainfall in the mid-Pliocene. Further, these rain amounts exceed the total annual rainfall in the present-day climate in some regions south of 20°S.

We find that only the PMIP4-CMIP6 model CESM2 simulates increased rainfall in Atacama region for the mid-Pliocene experiment, while the other models (IPSL-CM6A-LR, GISS-E2.1-G) simulate decreased rainfall. Simultaneously, CESM2 is the only model that realistically captures the sea-surface warming in the tropical and eastern Pacific. In the other GCMs the temperature increase in the mid-Pliocene is underestimated when compared to proxy data. However, it remains unclear whether the discrepancies in the simulated rainfall change patterns is caused by differences in the sea-surface temperature (SST). There is no clear improvement of tropical rainfall in tandem with reduced biases in SSTs in CMIP models (Fiedler et al., 2020). We cannot rule out that the representation of atmospheric circulation and clouds (Bony et al., 2015) contributes to differences in rainfall changes in the CMIP6 models.

Our regional evaluation is interesting in the context of the relatively high climate sensitivity of CESM2 (Gettelman et al., 2019), which might be seen as an outlier in a larger ensemble of CMIP6 simulations for other time periods (Burls and Sagoo, 2022). It was proposed to use paleo-simulations as testbed for climate model performance to constrain climate sensitivity (Burls and Sagoo, 2022, Zhu et al., 2022). Our results suggest that paleo-simulations paired with regional downscaling to kilometre-scales might also be useful for better understanding and predicting regional climate changes with global warming, e.g., for the hydrological cycle that remains an outstanding challenge for global models with parameterised convection. If our mid-Pliocene simulation is a useful out-of-sample test, the fact that CESM2 outperforms other models with lower climate sensitivity for the mid-Pliocene climate in the region of the Atacama Desert would support a high climate sensitivity. It would be valuable to have data from more global model simulation for the Pliocene or other warm climates for similar downscaling experiments in future research, especially from CMIP6 models with a high climate sensitivity. This endeavour requires also further development of proxy data for paleo climates, of which there are still a limited number for the Pliocene.

To the best of our knowledge this is the first time that a PMIP4 mid-Pliocene experiment has been downscaled with a regional climate model. We think that the unique datasets give new insights into the processes that are crucial for the mid-Pliocene climate in the Atacama Desert as an analogy for a warmer future world, and is valuable for the interpretation of paleo-climate proxies in the region, e.g., for understanding physical processes leading to different MCBs. One interesting aspect for future

research could stem from a detailed analysis of processes involved in some of the top mid-Pliocene rainfall events to gain a better understanding of the development of MCBs in the three degree warmer world of the mid-Pliocene, and putting it in the context of the El Niño Southern Oscillation (ENSO). Solman and Menendez (2001) state that El Niño events result in an equatorward shift of the winter storm track over the subtropical Pacific. Our results suggest that anomalously high SSTs might be beneficial for the formation of MCBs from the tropical Pacific. This implies that a Pliocene mean state more similar to today's El Niño conditions, like seen in some studies (e.g., Wara et al., 2005), may indeed contribute to wetter conditions in a warmer world via the processes identified in our study. However, the El-Nino like state of the Pliocene is subject of discussions (e.g., Fedorov et al., 2006; Watanabe et al., 2011; Oldeman et al., 2021), and hence the role of ENSO is not yet fully understood.

Atmospheric rivers are not unlike MCBs and are more strongly changing in the Northern Pacific than in the Southern Pacific between the mid-Pliocene and the pre-industrial (Menemenlis et al., 2021), which suggests that a dynamical assessment with high-spatial resolutions for the mid-Pliocene would also be interesting in other regions.

Furthermore, our findings have implications for rainfall extremes in a potential future '3-degree warmer' world, since the mid-Pliocene experiment is regarded as a potential analogue to what might be expected for the future. As such, our work suggests that regional rainfall extremes exceeding historic observations might occur. Such events involve atmospheric dynamics that are similar, but not identical to what we know from the recent past.

**Code availability**

The code for the regional climate model WRF is available online (https://www2.mmm.ucar.edu/wrf/users/download/get_source.html). The code to compute trough points can be inferred from Knippertz (2004). The code for the machine learning algorithms SOMs and k-Means are based on the Python libraries sompy.SOMFactory.build and Kmeans (sklearn.cluster).

**Data availability**

Global PMIP4-CMIP6 data is freely available via ESGF (e.g., https://esgf-node.llnl.gov/search/cmip6/). Rainfall for the Atacama Desert simulated with WRF using ERA-Interim as boundary conditions is freely available in the CRC1211 Database (doi: 10.5880/CRC1211DB.20). WRF simulated rainfall and moisture fluxes (IWVF) for the historical and mid-Pliocene experiment will be available when this paper is accepted for publication.

## Author contributions

MR is the main author and did main parts of analysis, conceptualisation, and writing. SF contributed to conceptualisation and writing, led the revision of the manuscript, and obtained the computing resources from DKRZ. PL contributed to the pre-processing of the CESM2 data, the WRF simulations, and writing. CB contributed to the analysis of the MCBs, conceptualisation, and writing. VW reviewed and interpreted the literature about mid-Pliocene proxy data and contributed to writing. YS contributed to conceptualisation and writing. SF and YS acquired funding and managed the project.

## Competing interests

The authors declare that they have no conflict of interest.

## Acknowledgements

This work used resources of the Deutsches Klimarechenzentrum (DKRZ, Hamburg) granted by its Scientific Steering Committee (WLA) under project ID bb1198. We gratefully acknowledge Gary Strand from NCAR for providing CESM2 data as boundary conditions for WRF. This research was funded by the Deutsche Forschungsgemeinschaft (DFG, German Research Foundation) – Projektnummer 268236062 – SFB1211 (https://sfb1211.uni-koeln.de/).

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

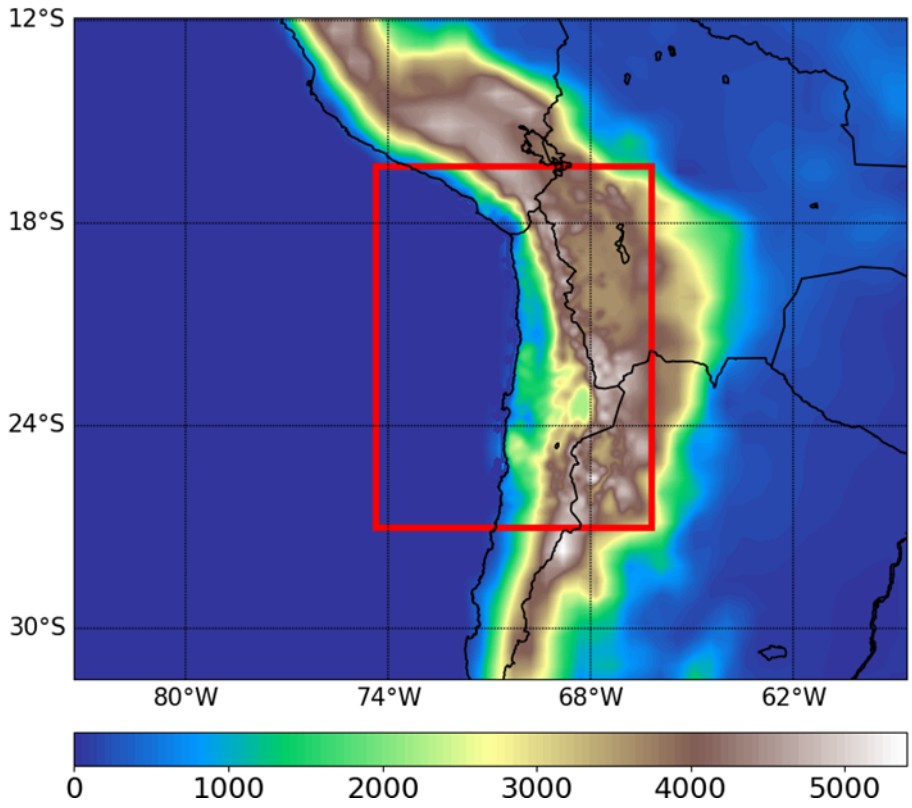

Figure 1: Map showing the outer (with 50 x 50km resolution) and inner WRF model domain (red box, with 10 x 10km resolution) and the topographic heights (shading, in m).

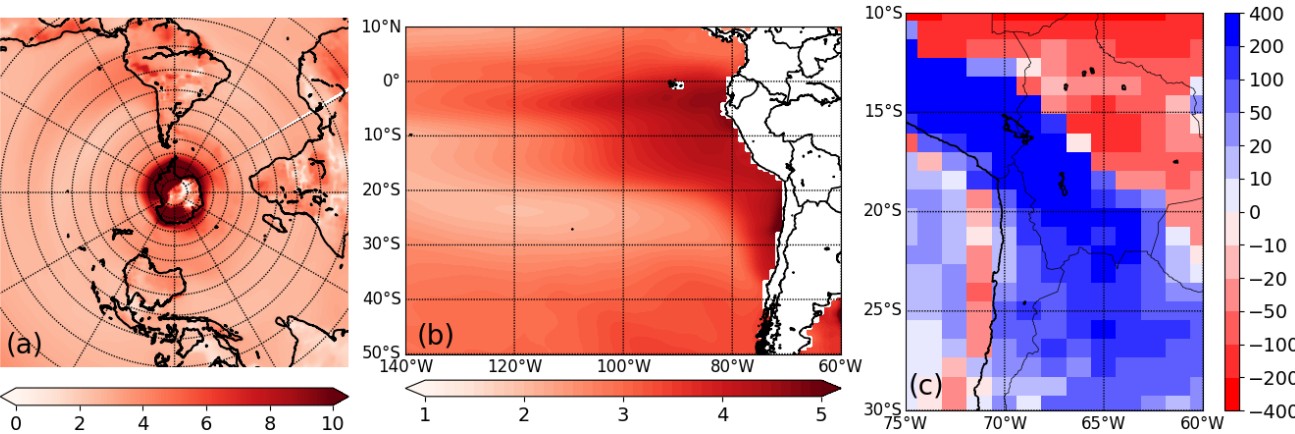

**Figure 2: Changes (mid-Pliocene minus historical) for mean annual (a) near-surface air temperature (at 2m above ground in °C), (b) sea-surface temperature (in °C), and (c) rainfall (in mm·yr⁻¹) as simulated by the global CESM2 model.**


'00

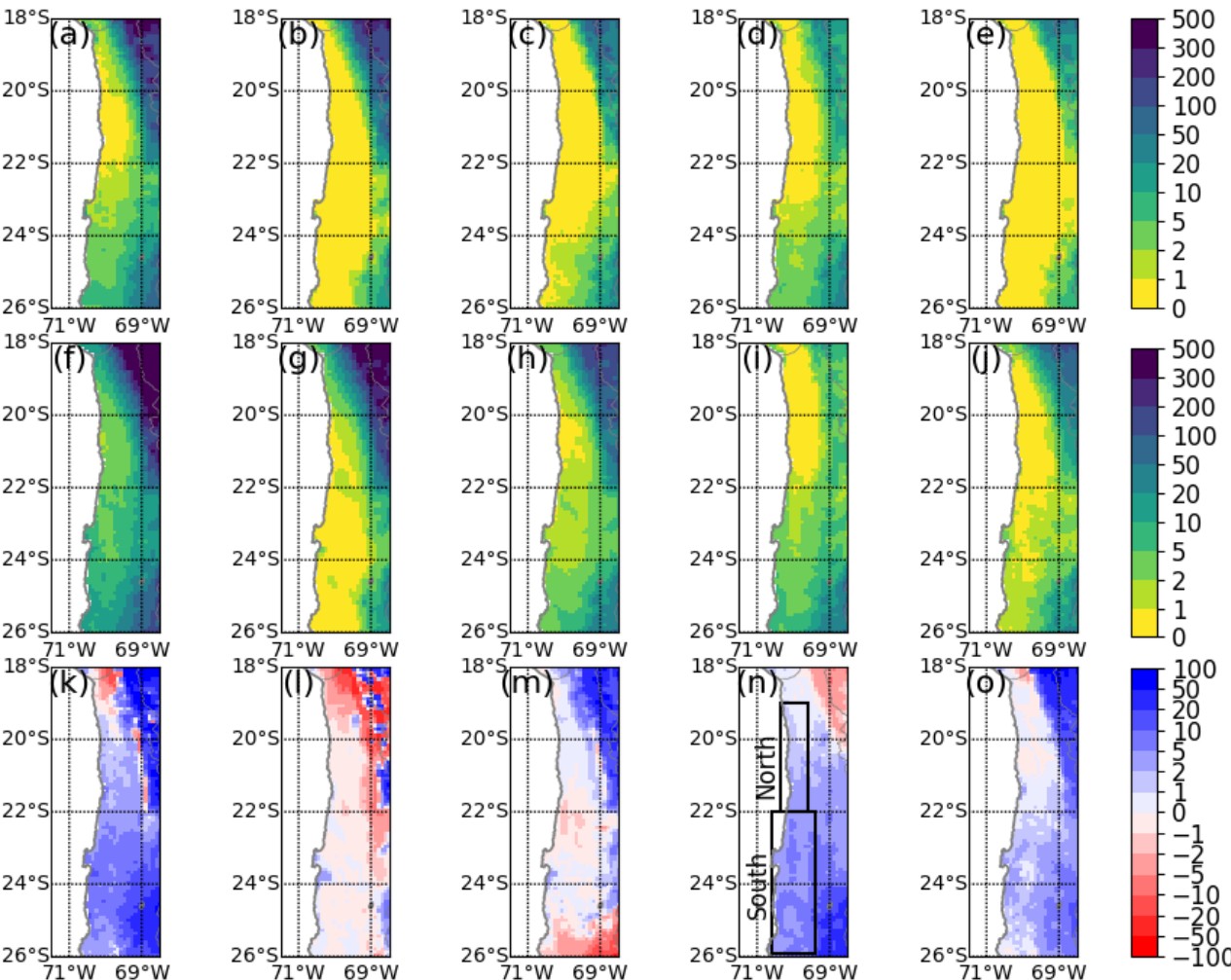

**Figure 3: Present-day mean annual and seasonal rainfall as simulated by (a-e) WRF$_{era}$ and (f-j) WRF$_{hist}$, and (k-o) mid-Pliocene changes (WRF$_{mP}$ minus WRF$_{hist}$) in the mean annual and seasonal rainfall. (a,f,k): annual; (b,g,l): DJF; (c,h,m): MAM; (d,i,n): JJA; (f,j,o): SON. All results are shown for the simulations with 10 km resolution, and values are given in mm·yr$^{-1}$ and mm·seas$^{-1}$, respectively.**

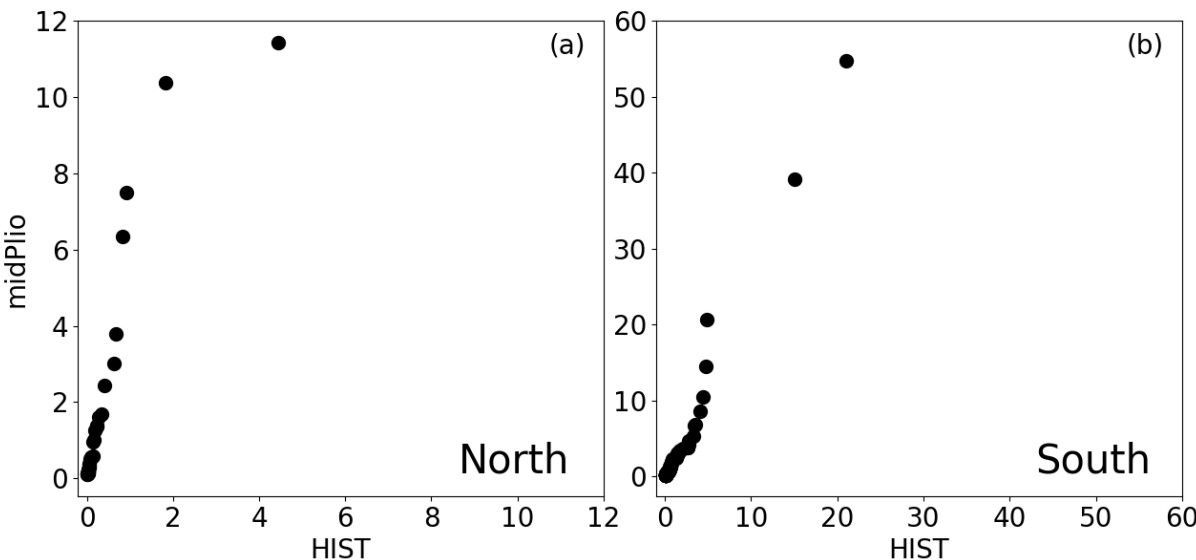

**Figure 4: Percentile-percentile plots for daily winter rainfall (in mm·day$^{-1}$) in WRF$_{mP}$ (y-axis) against WRF$_{hist}$ (x-axis) as spatial means over (a) the northern box and (b) the southern box, marked in Fig. 3n.**

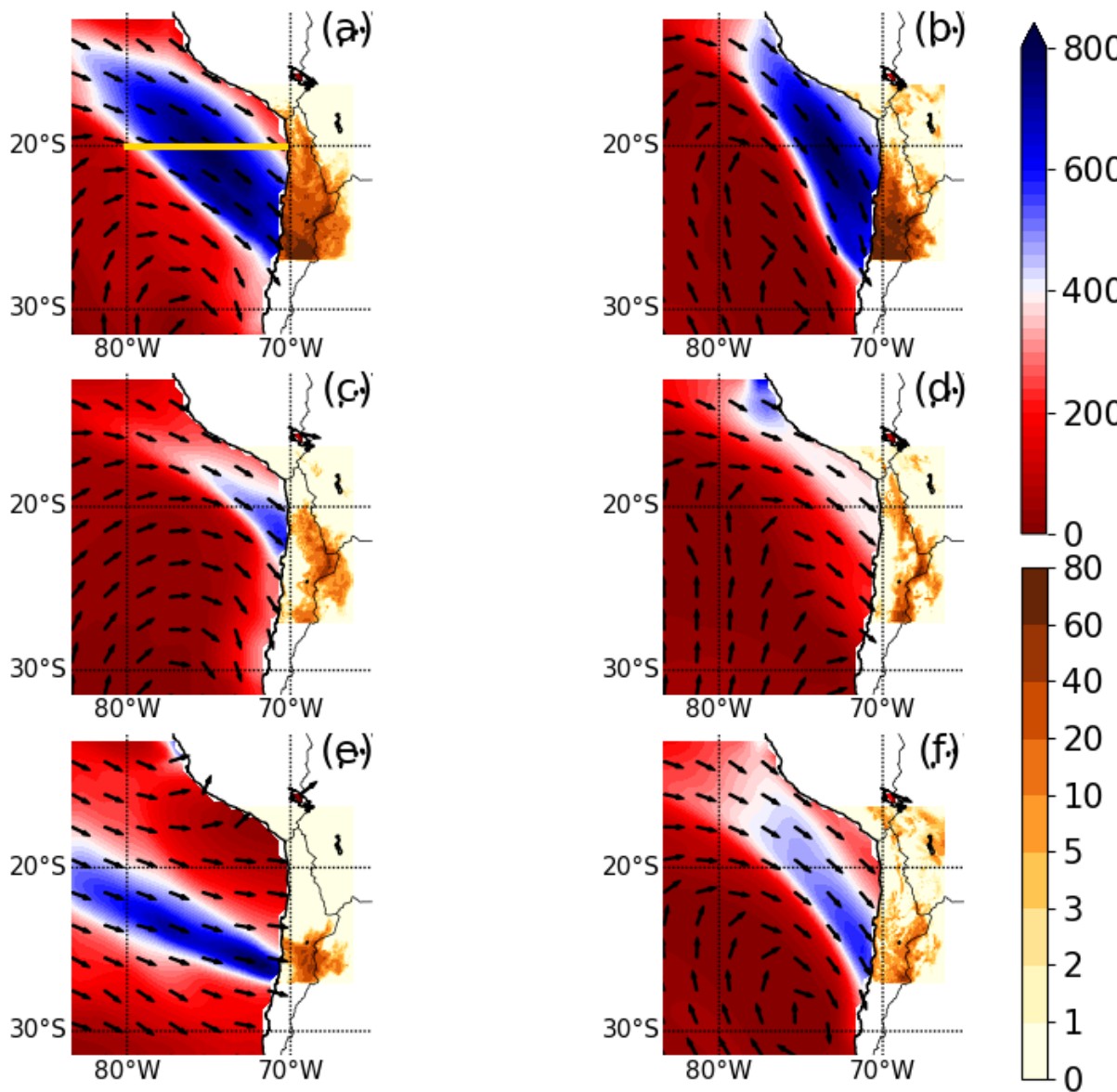

Figure 5: Daily rainfall over land (green-blue shading, in mm·day$^{-1}$) during top winter events in WRF$_{mP}$ (obtained for the inner model domain with 10 km resolution) and the maximum upper-level IWVF (above 2000m agl.) over ocean (arrows and red-blue shading, in kg·m$^{-1}$·s$^{-1}$) at the day of or the day prior the rainfall event (obtained from the outer model domain with 50 km resolution). (a,b) show events which are among the top events of both, northern and southern Atacama, (c,d) show events which are only among the top events of northern Atacama, and (e,f) show events which are only among the top events of southern Atacama.

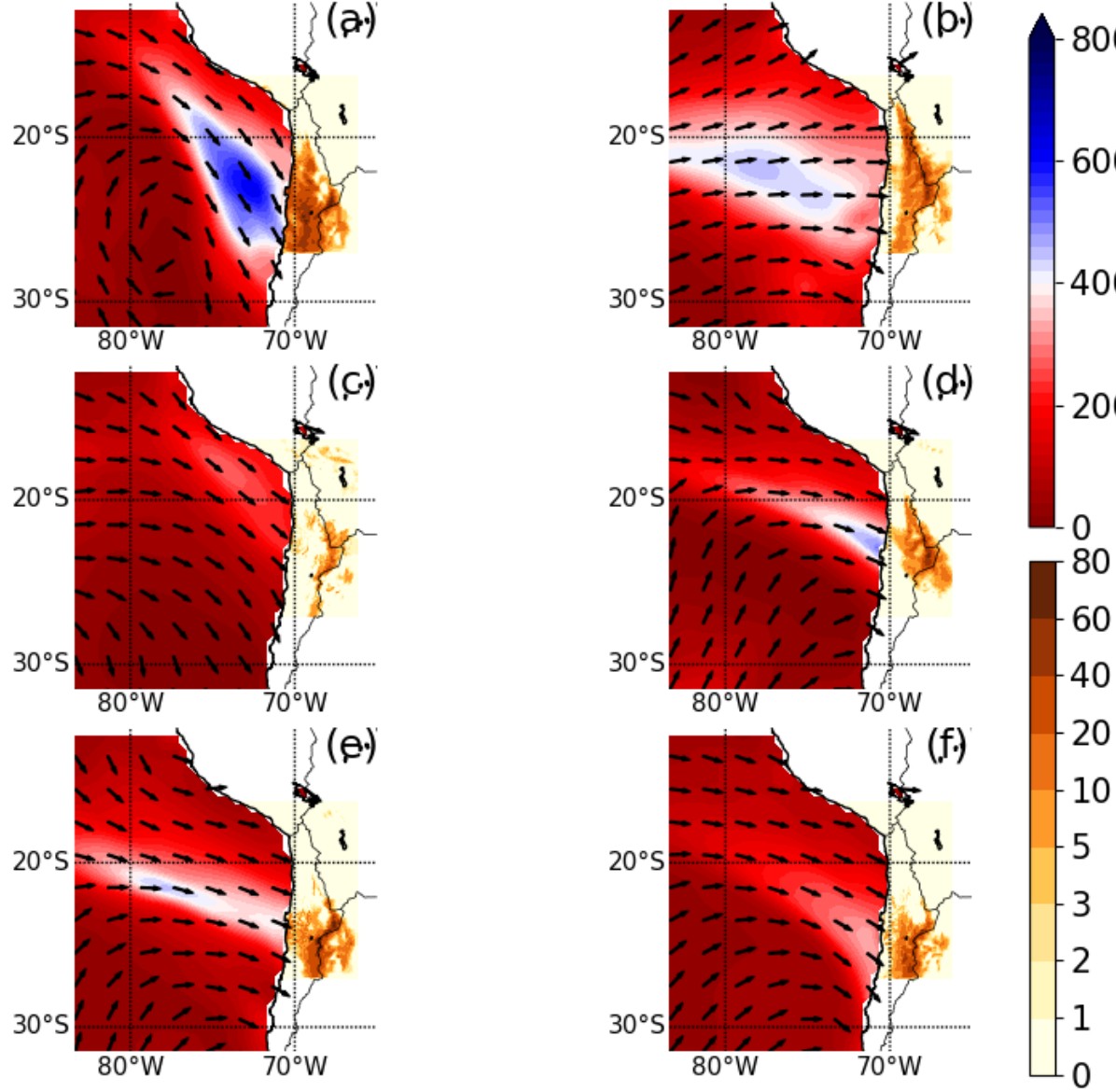

**Figure 6: As Fig. 5, but for the top winter events in WRF<sub>hist</sub>.**

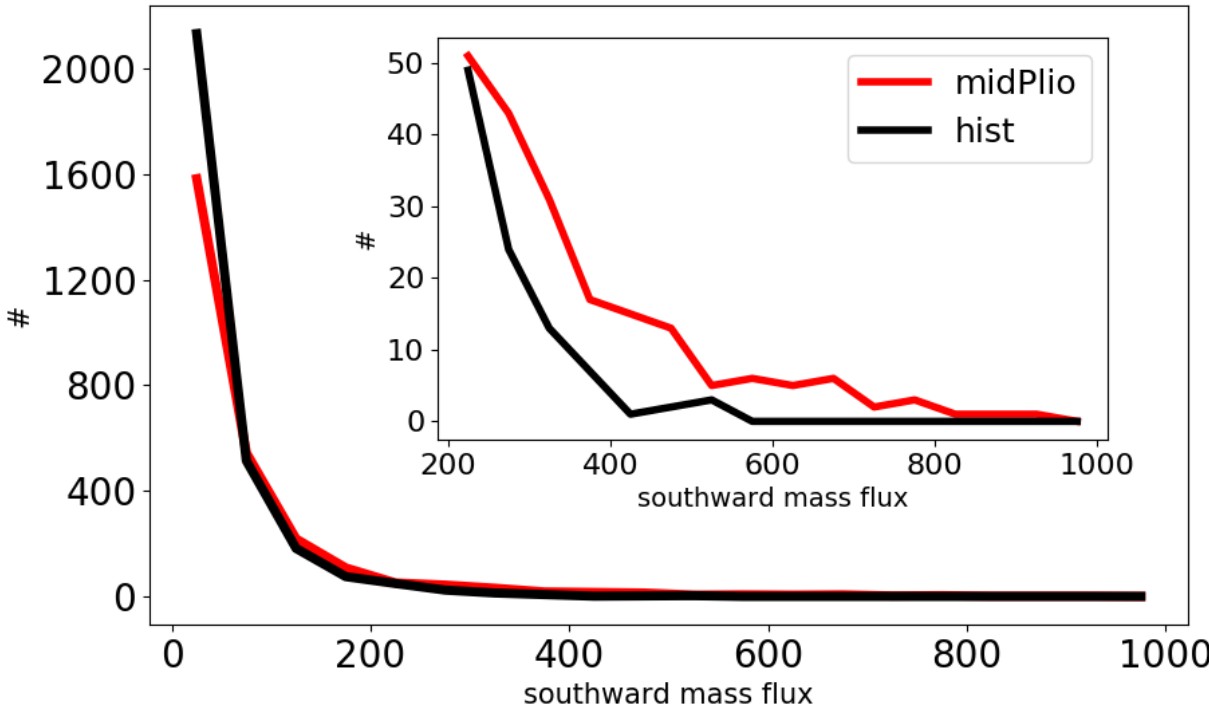

**Figure 7: Probability density function (y-axis, in total number # of dates) of 12-hourly southward upper-level (above 2000m agl.) IWV mass fluxes (x-axis, in $10^6 \cdot kg \cdot s^{-1}$) integrated between 80°W and 70°W at 20°S (see yellow line in Fig. 5a) in WRF$_{mP}$ and WRF$_{hist}$ (outer model domain with 50 km resolution) for Southern Hemisphere winter. The inset zooms into the results above $200 \cdot 10^6 \cdot kg \cdot s^{-1}$.**

50

55

60

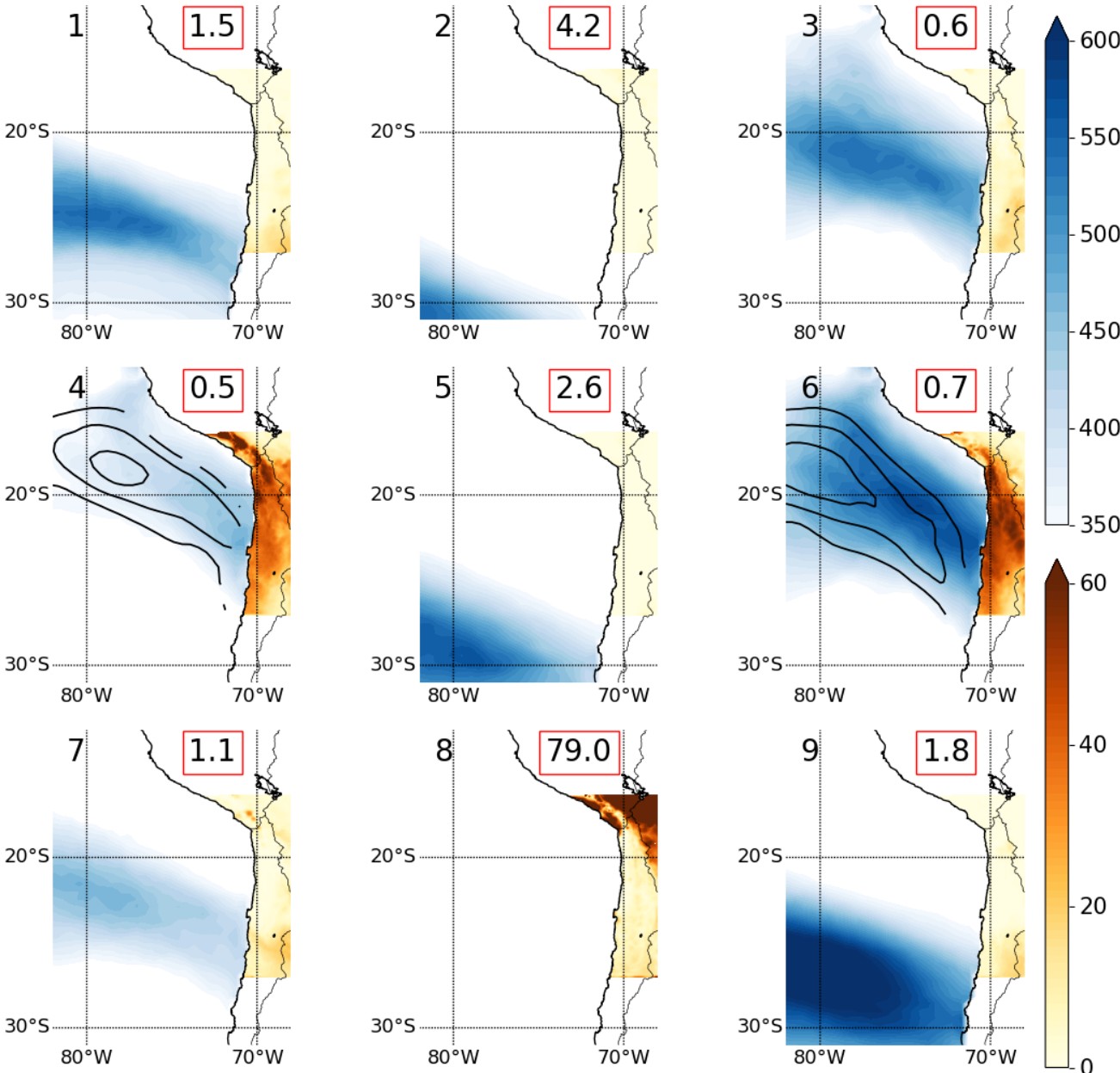

**Figure 8:** Final mid-Pliocene wintertime IWVF clusters (yellow-green shading, in kg·m⁻¹·s⁻¹) as obtained by the combined SOM and K-means clustering for WRF$_{mP}$ (outer model domain with 50 km resolution). The numbers in the red boxes are the frequencies of occurrence of the individual cluster (in days per Southern Hemisphere winter). Red-blue shading shows the fraction (in %) of the rainfall associated with the individual clusters to the total winter rainfall (inner model domain with 10 km resolution). The black contours in cluster 4 and 6 show the composite mean wind speeds in 4000 m asl. (15, 17, and 19 m·s⁻¹). Note that we show IWVF exceeding 350 kg·m⁻¹·s⁻¹ in accordance with the definition of MCBs for the mid-Pliocene (see Methods).

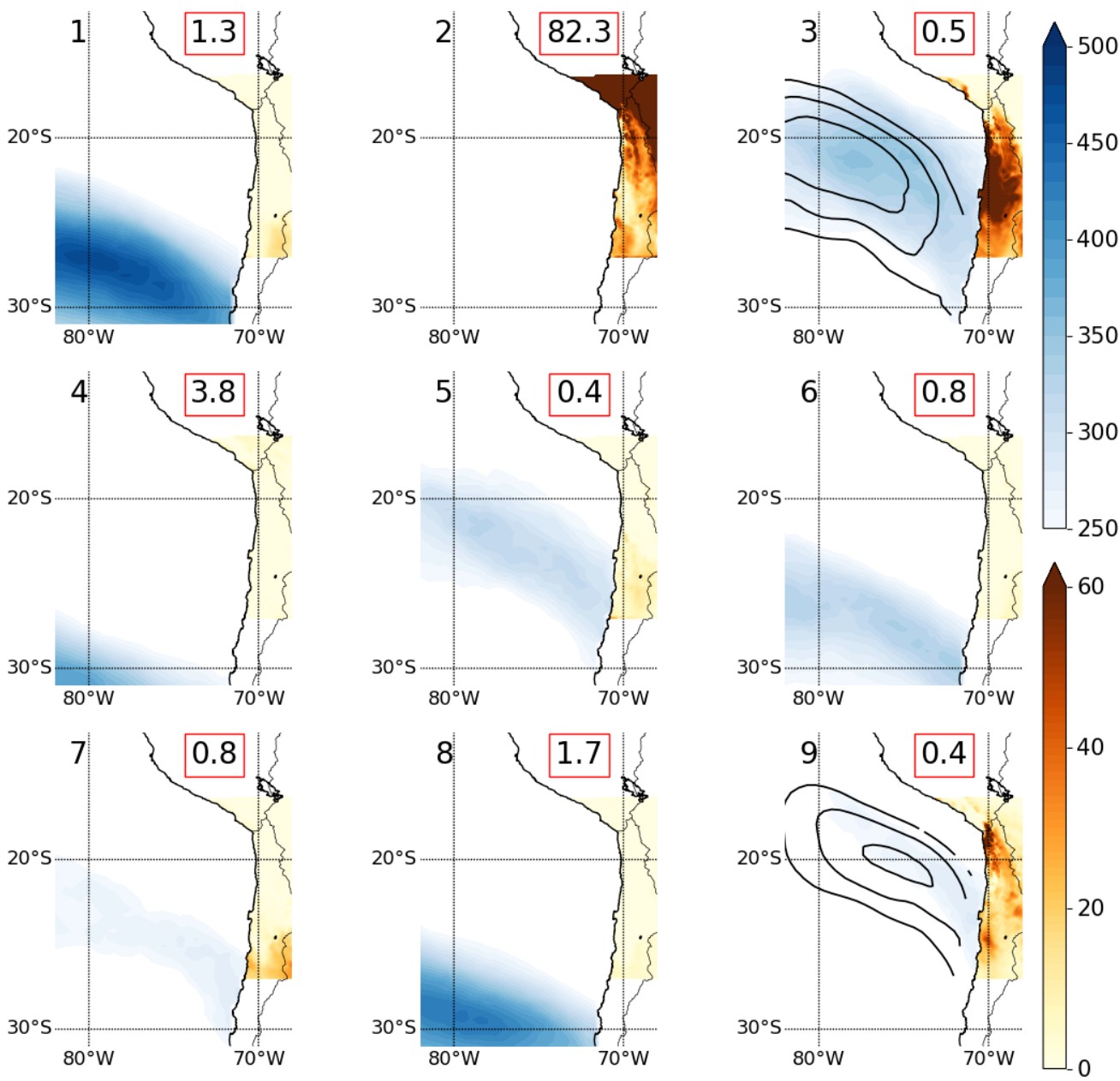

**Figure 9: As Fig. 8, but for WRF$_{hist}$. The black contours in cluster 3 and 9 show the composite mean wind speed in 4000 m asl. (15, 17, and 19 m·s$^{-1}$).**

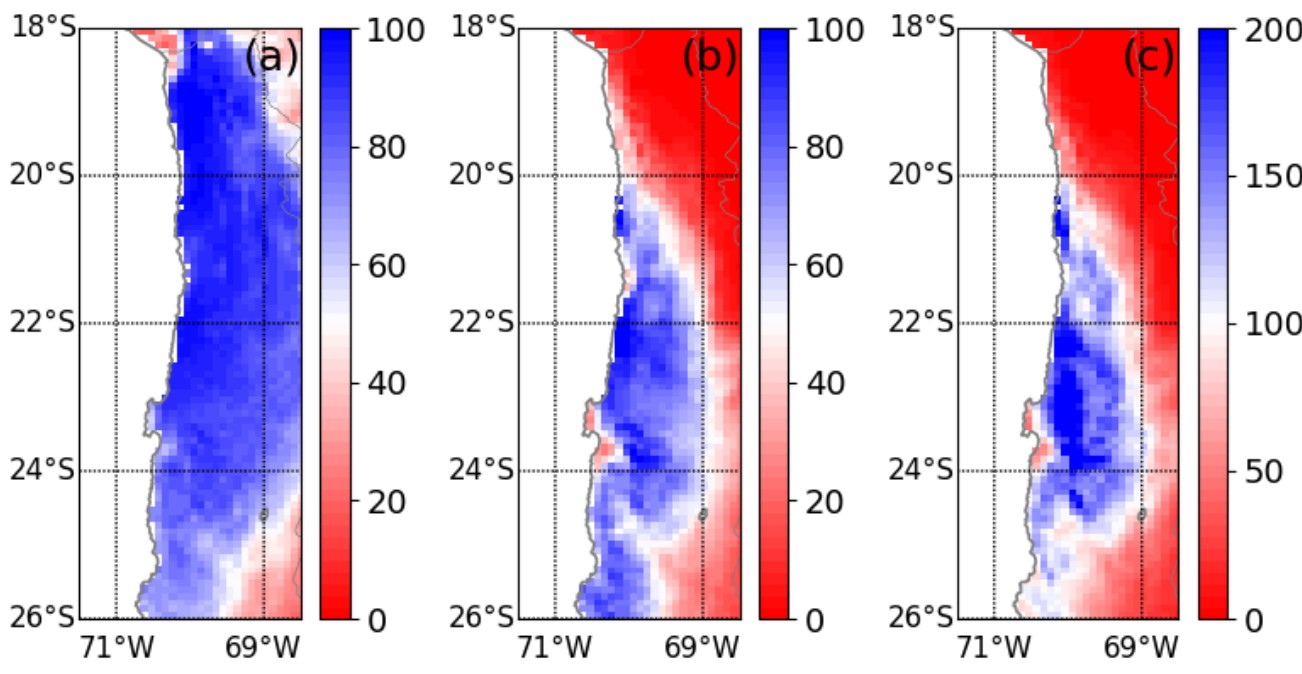

**Figure 10: Fraction (in %) of the rainfall associated with MCB cluster 4 and 6 of WRF$_{mP}$ to (a) total winter rainfall in WRF$_{mP}$, (b) total annual rainfall in WRF$_{mP}$, and to (c) total annual rainfall in WRF$_{hist}$ (inner model domain with 10 km resolution).**

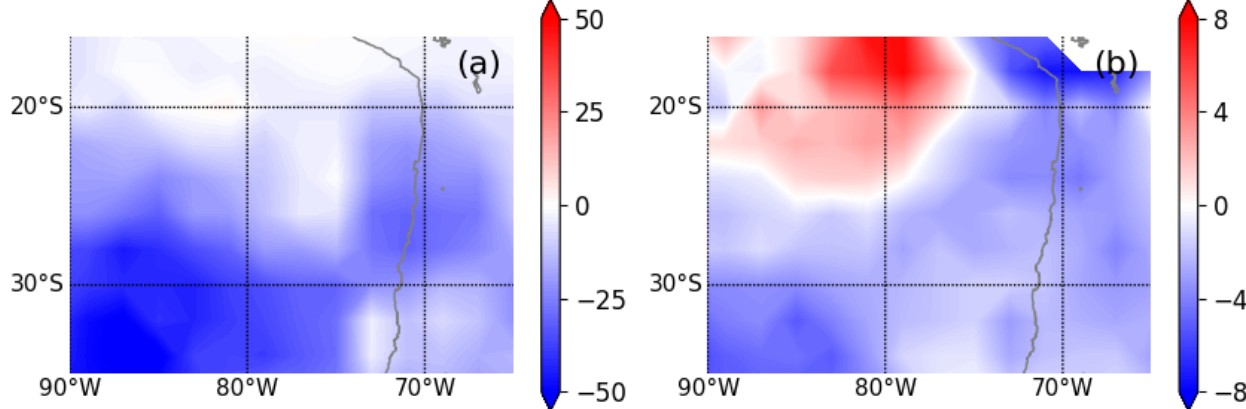

**Figure 11: CESM2 mid-Pliocene minus CESM2 historical for (a) number of troughpoints (in number per decade), and (b) mean troughpoint strength (in gpm).**

| Name of site | Coordinates | Time period | Type of proxy data | Signal relative to present-day conditions | Reference |
|---|---|---|---|---|---|
| Cerro Soledad, Quillagua-Llamara basin | 21.25° S; 69.5° W | 3.2–2.7 Ma | Cosmogenic nuclide dating of lake terraces | Wetter conditions in the Altiplano | Ritter et al. (2018) |
| Soledad Fm, Quillagua-Llamara basin | 20-21° S; 69-70° W | 4.2-2.6 Ma | Ash layers in playa-lake sediments | Wetter conditions in the Altiplano | Vásquez et al. (2018) |
| Tiliviche Paleolake | 19.5° S; 70° W | 3.5-~3.0 Ma | Salar deposits in the Tivliche paleolake | Wetter conditions in the Altiplano | Kirk-Lawlor et al. (2013) |
| Lauca basin | 18.5° S 69.25° W | 3.7–2.6 Ma | Lacustrine and fluvial sediments | Local proxy for semi-arid conditions with increased precipitation | Gaupp et al. (1999) |
| Cordillera de la Sal, Salar de Atacama basin | 23° S 68.25° W | 3.5 – 2 Ma | Lacustrine and mudflat deposits | Wetter conditions in the Cordillera | Evenstar et al. (2016) |
| Calama Basin | 22.5° S 69° W | 6 – 3 Ma | Palustrine carbonates | Wetter conditions in the Altiplano | May et al. (2005) |
| Central Depression, Calama basin, and Preandean Depression | 19.75 −23° S | 8 – 3 Ma | Fluviolacustrine and alluvial-fan deposits | Semi-arid conditions | Hartley & Chong (2002) |
| Coastal Cordillera draianges | 23.45 - 29.9° S | > 2.1 Ma | Cosmogenic nuclide dating and near-surface ash ages | Wetter conditions | Amundson et al. (2012) |

Table 1: Proxy data for wetter condition in the region of the Atacama Desert that fall into the mid-Pliocene.

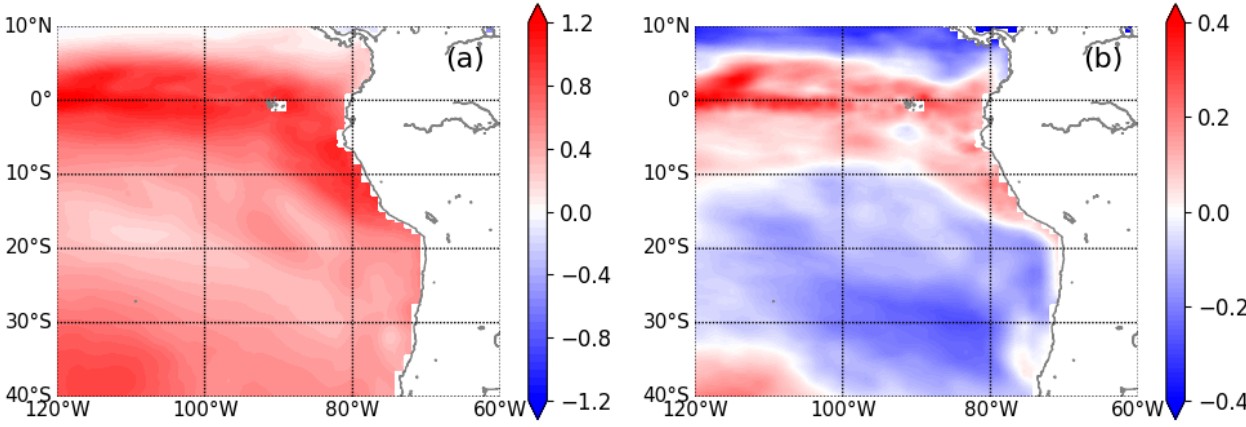

**Figure 12: SST composite mean over all days with MCB cluster 4 and 6 minus (a) 50th percentile of winter SST and (b) 70th percentile of winter SST in the mid-Pliocene experiment of CESM2.**

