# Peer review of "On the importance of moisture conveyor belts from the tropical East Pacific for wetter conditions in the Atacama Desert during the Mid-Pliocene"

_Climate of the Past, 2022_

## Author Comment (AC1)

**Point-by-point reply for Reyers et al.:**
**„On the importance of moisture conveyor belts from the tropical East Pacific for wetter conditions in the Atacama Desert during the Mid-Pliocene"**

We thank all community members who provided comments on our manuscript for the appraisal of our manuscript. The comments helped us to further improve the presentation of the results in the manuscript. Our replies to the comments along with details on how we intend to revise the manuscript are printed in blue below the original comments in black. We also revise the color schemes in the figures for clarity.

**Reply to CC1 by Arthur Oldeman**

„Dear authors, You submitted a nice work to CP, advancing the science on the mid-Pliocene hydrological cycle. So well done! I do think that your work misses some key background information on the mid-Pliocene (Pacific) hydrological cycle, based on recent PlioMIP2 publications.
•        The mid-Pliocene simulations of CESM2 you are using are further analysed in Feng et al 2020. They specifically also look at the tropical Pacific circulation changes, so Hadley & Walker circulation and equatorial Pacific SSTs. https://doi.org/10.1029/2019MS002033
•        An evaluation of the large scale hydrological cycle response in PlioMIP2 (including the CESM2 simulations) is included in Han et al 2021. They find wetter conditions in the deep tropics, so Pacific ITCZ, and give an explanation for where this moisture is coming from. https://doi.org/10.5194/cp-17-2537-2021
•        Pontes et al 2020 research PlioMIP1 and PlioMIP2 results and find a northward shift of the ITCZ and a weakened and poleward displaced South Pacific Convergence Zone (SPCZ). These results might again be relevant for where exactly the moisture in the atmosphere is coming from in your region of interest. https://doi.org/10.1038/s41598-020-68884-5
•        Related but maybe less relevant is Gabriel Pontes' recent publication, relating the northward shift of the Pacific ITCZ to the reduced El Nino variability in the PlioMIP1 and 2. https://doi.org/10.1038/s41561-022-00999-y
•        A very relevant study, using PlioMIP2 model CCSM4-UoT (so not CESM2) is Menemenlis et al 2021, where they study and attribute precipitation changes in the mid-Pliocene, a.o. focusing on the coastal area of Chile. They explain & attribute the precipitation changes to dynamical changes in atmospheric rivers. https://doi.org/10.1016/j.gloplacha.2021.103557
I saw you do include my 2021 paper on reduced El Nino variability in the PlioMIP2 ensemble (Oldeman et al). I think that most papers I refer to above are actually more relevant to your study than that article, since these focus more on atmospheric dynamics and the hydrological cycle rather than SST variability.
Your work is interesting but lacks - in my view - some relevant background knowledge on the hydrological cycle and atmospheric dynamics from modelling studies. I would recommend including some of the content of these publications either in the Introduction, or section 3.2 Potential drivers.
Technical comment: you consistently refer to "PlioMIP" in the Methods section. CESM2 was included in phase 2, so PlioMIP2. It would be good for findability to use PlioMIP2 (just as you are using CMIP6 and PMIP4 rather than CMIP or PMIP).
Much of luck with the research!
Best regards, Arthur Oldeman"

Dear Arthur Oldeman,

thank you very much for your interest in our manuscript. We appreciate your suggestions for further links to published works and include citations to articles that you suggest in our revised manuscript.

Specifically, we add in the introduction: "For the warmer mid-Pliocene climate, the multi-model mean of the PlioMIP2 models for instance indicate that the Hadley Cell was shifted northward and the Walker Circulation shifted westward (Han et al., 2021).", and in Section 3.2: "Thermodynamic and dynamical changes in PlioMIP2 models lead to wetter tropics, particularly in the eastern tropical Pacific (Han et al., 2021). Furthermore, the subtropical anticyclones over the southern Pacific intensified and expanded further to the West in PlioMIP2 (Pontes et al., 2020). The change in the synoptic-scale circulation is a potential reason for the smaller frequency of troughs reaching the subtropical Pacific offshore the Atacama (Fig. 11a).",

The studies with inter-comparison results are cited in the method section: "(…) the validation results of GCMs participating in PlioMIP2, assessed previously (e.g., Feng et al., 2020, Pontes et al., 2020, Han et al., 2021)."

We add in the conclusions: "Atmospheric rivers are not unlike MCBs and are more strongly changing in the Northern Pacific than in the Southern Pacific between the mid-Pliocene and the pre-industrial (Menemenlis et al., 2021), which suggests that a dynamical assessment with high-spatial resolutions for the mid-Pliocene would also be interesting in other regions."

For consistency, we use the term PlioMIP2 instead of PlioMIP throughout the revised manuscript.

---

## Author Comment (AC2)

**Point-by-point reply for Reyers et al.:
„On the importance of moisture conveyor belts from the tropical East Pacific for wetter conditions in the Atacama Desert during the Mid-Pliocene"**

We thank all community members who provided comments on our manuscript for the appraisal of our manuscript. The comments helped us to further improve the presentation of the results in the manuscript. Our replies to the comments along with details on how we intend to revise the manuscript are printed in blue below the original comments in black. We also revise the color schemes in the figures for clarity.

**Reply to RC1 by Teresa Jordan**

„Overall, this is an excellent contribution to paleoclimate studies of the Atacama desert of the west coast of South America, as well as a being a thought-provoking treatment of what future extreme events may arise in the Atacama Desert during a warmer climate future.
The largest excursion from hyperaridity in the Atacama Desert more recently than about 10 Ma was during the Pliocene, and this is the focus of Reyers et al. atmospheric modeling study. The proxy record has not firmly established most of the key parameters of the Pliocene wetter climate – was it only slightly wetter but over a very long period of time (e.g., 500,000 years) allow accumulated impacts on the landscape, or was it intensely wetter for some shorter period of time (e.g., 1000 years) which caused rapid changes of landscape features? Nor is it firmly established the timing of the wettest interval. Some data point to the wettest interval in the 20°S Atacama at the very beginning of the Pliocene (e.g., Jordan et al., 2014; Evenstar et al. 2017), thus not coinciding with the mid-Pliocene boundary conditions used by Reyers et al. (their conditions are appropriate to 3.2 Ma. Nevertheless, given the dearth of rigorous atmospheric studies that attempt to evaluate the warmer conditions of the Pliocene, I find Reyers et al's analyses to be both novel and important. In light of the lack of detailed proxy understanding of the Pliocene wetter climate extracted thus far from the geological record, perhaps the authors should state in their conclusions that the applicability to interpreting paleoclimate history also requires improved chronological resolution and more advanced proxy data."

Dear Teresa Jordan,

thank you very much for your evaluation of our study. We included the following in the revised conclusion: „This endeavour requires also further development of proxy data for paleo climates, of which there are still a limited number for the Pliocene.".

„Read as a person who is not an atmospheric scientist, I find the premises and methods to be clearly stated, and the nature of the experiments to be very good choices for both Pliocene conditions and with reference to the March 2015 extreme rain event case study. With only minor exceptions, the results presented support well the interpretations and conclusions.
The authors 'choices of features to illustrate in figures and the simple clarity of the illustrations are very good. The connection between topics in the text and corresponding figure was easy to match.
The choices of materials to reference is suitable. In a few cases in which I know well the paper that is cited I recommend that the authors re-examine the paper cited and more clearly (or accurately) represent its conclusions."

Thank you, we carefully revise the manuscript text for clarity, e.g., by adding more details for citations and interpretations. Please refer to our replies below for more details.

„Abstract: Easy to understand
Data and Methods, section 2:

The authors describe very clearly their analysis and machine learning methods. These topics are entirely outside of my expertise, yet I could follow the description easily.
Introduction: This frames the problem well, from a geological perspective."

Thank you for your feedback.

„A sentence spanning lines 54- 57 describes a prior interpretation that atmospheric circulation over South America east of the Andes controlled increased precipitation in the Atacama in the past, ie, paleoclimate. The citation of references is ambiguous. On first reading, it appeared that Jordan et al. (2019) and Amidon et al. (2017) both advocated this interpretation. However, Jordan et al described the previous literature on this topic but concluded that it is incorrect for the situations in the Atacama Desert of interest in their paper. (Amidon et al's paper focus on the eastern flank of the Andes, where perhaps the interpretation is correct.) While the authors of that paper are pleased that their description could be understood and contributed to thought, the citation should be changed to make clear that they conclude it to be incorrect."

The discussion of the topic in the cited papers were useful. We revise the text for clarity: "Past rainfall variations in the Northern Atacama and the Andes have been linked to latitudinal shifts of the extra-tropical westerlies in the Southern Hemisphere (Amidon et al., 2017). Also, cut-off lows as seen in March 2015 have been proposed as possible mechanism for wetter conditions in the past (Jordan et al., 2019)."

„A sentence spanning lines 66 to 68 treats the March 2015 extreme rain event and cites Jordan et al (2019) for relating that tropical Pacific-sourced event to a paleoclimate hypothesis. Because Jordan et al (2019) did not present isotopic data for paleoclimate proxies, the citation is somewhat distorted. Were a citation to be added to either Herrera and Custodio (2014) or Herrera et al. (2018), then the paleoclimate proxy isotope data would be encompassed and Reyers et al. intent for the sentence could be better fulfilled.
Herrera, C., and Custodio, E., 2014, Origin of waters from small springs located at the northern coast of Chile, in the vicinity of Antofagasta: Andean Geol, v. 41, p. 314–341.
Herrera, C., Gamboa, C., Custodio, E., Jordan, T., Godfrey, L., Jódar, J., Luque, J.A., Vargas, J., and Sáez, A., 2018, Groundwater origin and recharge in the hyperarid Cordillera de la Costa, Atacama Desert, northern Chile: Science of The Total Environment, v. 624, p. 114–132."

We change the sentence for clarity: „Based on the characteristic isotopic composition of rain water  (e.g., Herrera and Custodio, 2014) from the March 2015 event, Jordan et al. (2019) proposed that the processes involved in this event might also play an important role in increased paleoclimate rainfall in the Atacama Desert"

„Line 79: Please remove the citation of Jordan et al. (2019). That paper advocates that the tropical Pacific is a key contributor to moisture in the Atacama Desert, and only describes the sources promoted by the other authors for completeness. Jordan et al's analysis aligns better with Böhm et al 2021 than with the other papers listed, and could be cited at the end of the sentence which terminates in the middle of Line 80."

We revise it as suggested: „ If MCBs played an important role in that period, this would imply that in addition to the previously suggested regions southwest or east of the Atacama Desert (Stuut and Lamy, 2017; Bartz et al., 2019; Amidon et al., 2017) also the tropical Southeast Pacific northwest of the desert could be a potential moisture source for increased humidity in the mid-Pliocene, like assessments of the regional rainfall under present-day climate suggest (Bozkurt et al., 2016, Jordan et al., 2019; Böhm et al., 2021). "

„Line 95: One of the two words with the same root, "description" and "described" should be removed, and the sentence slightly rewritten."

Agreed, now: „Dowsett et al. (2016) describes PRISM4 in detail. "

„Line 121. I believe this is the first call for Fig. 1. The caption for Fig. 1 should specific what "surface temperature" refers to. Sure, among atmospheric scientists perhaps it is understood that this means "air temperature 1 m above the surface, irrespective of whether the surface is water or soil". But the paleoclimate-geologist does not know (and may have guessed incorrectly.)"

Figure 1 shows the topography and is first referred to in the introduction, which is now more explicit: „(…) shown in Fig. 1."
We added the details for the temperature in the caption of Figure 2: „ (…) (a) near-surface air temperature (at 2m above ground in °C), (b) sea-surface temperature (in °C) (…)"

„Results
Section 3.1: Mid-Pliocene against present-day climate
Line 177. It seems appropriate to also refer to empirical data for the magnitude of temperature anomaly during the mid-Pliocene."

Yes, we add: "The model results are supported by proxy data indicating a global SST anomaly for the mid-Pliocene vs. pre-industrial of 2.3°C and 3.2–3.4°C based on foraminifera Mg/Ca and alkenones or alkenones only, respectively (McClymont et al., 2020). Specifically in the upwelling regions at the Peruvian margin, Deckens et al. (2007) reconstructed a Pliocene-modern SST change by 2.9°C"

„At the end of this section (Lines 209-210) the Atacama is split into two sectors, N and S, for further analysis. The choice of boundary between the two sections and the E-W dimensions of the sectors corresponds well to both modern and paleo-climate subdivisions."

The choice of the subdivision is motivated by the different seasonal cycles of rainfall between the northern and southern part of the hyperarid Atacama region. We add this information in the manuscript: "The seasonal cycle of rainfall shifts from winter-dominated in the south to summer-dominated in the north of the Atacama Desert (Houston, 2006; Reyers et al., 2020). In order to account for these differences, we split the hyper-arid core into a northern and southern region (see black boxes in Fig. 3n), which has also been done similarly in previous studies (e.g. Böhm et al., 2020). To quantify the cause of the winter rainfall increase (see Fig. 3n), we compute the spatial means of daily rainfall over these two regions."

"Line 210-211. A phrase should be added to remind the reader what are the "things" of which there are 15 in both the mid Pliocene and historical simulations (this is the number of symbols). It is clearly stated that each "thing" is a spatial mean. However, it is not stated whether those 15 "things" are computational repetitions of the same model (like a Monte Carlo simulation) or something else (is 15 the maximum number of days it rained?). Perhaps this was explained in the Data and Methods sections, but the reader needs clarification at this point in the text."

Agreed, we change it to: „Then we rank the spatially averaged daily rainfall amounts according to their magnitude and display the results in percentile-percentile plots for the mid-Pliocene against the present-day simulation (Fig. 4)". There is one dot for each value in the data set. There are many overlying dots close to zero, but these appear as just one dot in the chosen figure type and therefore it seems as if there are around 15 dots.

„Section 3.2; Potential drivers for stronger rainfall events in the mid-Pliocene
Figure 4: It would be slightly easier for the reader if, within the boxes that contain (a) and (b), the words "north" and "south" occur. Yes, that information is in the caption. But for the person glancing repeatedly at the figure while reading the text, this added label would be advantageous."

We add the words in the figure.

Line 242: The text should refer specifically to the inset within Fig. 7. The feature emphasized in this and the next sentences cannot be seen in the large-area plot.

Agreed, it is now: „The extremely strong southward mass fluxes are more frequent in $WRF_{mP}$ than in $WRF_{hist}$ (inset in Fig. 7)."

Line 250 and caption for Figure 8: The first appearance of the word "cluster" in the sentence (line 250) and caption should be plural (i.e., "clusters").

Thanks, we correct it.

Figure 8. It seems peculiar that the most common cluster, with 79 occurrences per winter (cluster 8), has 0 kg/m-s of water vapor. I deduce this from the lack of any color on the yellow-green scale. Has color been accidently omitted? Perhaps the color bar needs a different stretch so that a non-zero value is visible? If 0 is the correct interpretation of Fig. 8 cluster 8, then the caption and text ought to mention this. At present, Line 252 discusses this cluster with reference to easterly IWVF, which I do not see.

There is no sufficiently strong moisture advection from the West in cluster 8 to fulfil the criterion of a MCB. The IWVF is not zero, but below the chosen threshold of 350 kg $m^{-1}$ $s^{-1}$ for MCBs. The moisture for the rainfall in the North of the Atacama Desert rather stems from Easterly directions, described in lines 250-253 with reference to Reyers et al. (2021) for further details. We add in the caption: „ Note that we show IWVF exceeding 350 kg·$m^{-1}$·$s^{-1}$ in accordance with the definition of MCBs for the mid-Pliocene (see Methods). "

Line 254. Clusters 1, 2, 5, 7, 9 for the mid-Pliocene are described as similar. To my eye, 7 does not belong in this group. Its IWVF looks more like 3 or 4. This leaves the impression that the designation of "zonal" is quite arbitrary. It would be appropriate to describe the differences in the systems responsible for clusters 3, 4, and 7 (e.g, was it SST that differences? Or wind strength?).

We give more details for clarifying why we think these clusters are similar to each other in the revised text:
„ Cluster 1, 2, 5, 7, and 9 include more zonally oriented IWVF with landfall in central or southern Chile with origins further towards the South of the Pacific, e.g., when compared to cluster 4. Also cluster 3 is associated with moisture advection from regions South of 20°S but includes also some transport from further North with a more tilted axis towards meridional directions compared to cluster 1, 2, 5, 7, and 9. These clusters with dominant moisture advection from regions South of 20°S play a minor role for winter rainfall in the Atacama Desert (see red-blue shading in Fig. 8). In contrast, the IWVF clusters 4 and 6, which occur approximately every second winter in $WRF_{mP}$, produce the largest rain amount during winter in the hyper-arid core of the Atacama Desert. " We focus on the clusters 4 and 6 that lead to rainfall events in the Atacama Desert, since we aim to explain why the Pliocene Atacama was less arid, but the moisture uptake paired with the circulation patterns prevent rainfall in the desert in cluster 3 and 7. We also add in the conclusion: „ (…) e.g., for understanding physical processes leading to different MCBs. "

„Line 258-260. Clusters 1, 4, 6, 7, 8 for Historical are described as similar. The text speaks of "landfall south of 25°S". Was that also the criteria used when comparing the Mid-Pliocene clusters, but was not stated?"

We used the dominant origin of moisture paired with rainfall activity in the core of the Atacama as criteria. Please see reply to previous comment.

„Line 263. The text should note that the "magnitude" considered is integrated water vapor."

Revised to: „(…) have much smaller IWV (…)"

„Line 263. The text states that the Historical MCBs that provide most of the rain (Fig. 9, clusters 3 and 9) are shifted south relative to those of the mid-Pliocene (Fig. 8, clusters 4 and 6). I do not see a significant latitudinal difference. The long axes of the mid-Pliocene clusters cross the 80°W meridian at about 18 and 19°S; the long axes of Historical clusters cross 80°W at 21°S and 17°S. If this is significantly different, the authors need to demonstrate the significance statistically."

The change in latitude is less apparent than for the magnitude of IWV. We revised the text to make the differences clearer: „However, compared to the MCBs of $WRF_{mP}$ (Fig. 8), the MCBs in $WRF_{hist}$ (in Fig. 9) have much smaller IWV and can have different transport characteristics, e.g., indicated by wind maxima that are shifted towards the South and East. Take for instance, the smaller IWV and different wind pattern in cluster 4 (6) for $WRF_{mP}$ in Fig. 8 against cluster 9 (3) for $WRF_{hist}$ in Fig. 9."

„Lines 264-265. It is stated that the MCBs of the mid-Pliocene are of clearly different origins than present-day. Clarification is needed of this statement. It seems to me that a few degrees of latitude would not constitute a "different origin" unless those few degrees place the air mass transport path over significantly different parts of the ocean, for instance, markedly different SST. If there is an important difference of this type, the authors should state this as justification of the statement that the origins are "clearly different.""

We have removed the word „clearly". Please also see our reply to the previous comment.

„Line 268. The phrasing is unclear. I believe it would be correct if written "Examples of the wind fields at 4000 m asl. are shown in Fig. 8 and 9.""

We revise the text for clarity: „The strengthening of the north-westerly MCBs in $WRF_{mP}$ might suggest that the wind speeds associated with these MCBs are higher in the mid-Pliocene leading to rainfall on land, but this is not visible in the output. We computed the composite of mean wind speeds for each of the MCB clusters associated with rain in the desert. The resulting composite of wind speeds in the clusters are marked for a typical moisture transport height of 4000m asl. in Fig. 8 and 9. Interestingly, the mean wind speeds in the two clusters in $WRF_{mP}$ have a similar magnitude like in $WRF_{hist}$. However, the region with peak winds is shifted to the north-west in $WRF_{mP}$. We therefore conclude that the MCBs during the mid-Pliocene that lead to rain in the Atacama are not only stronger, but in some cases also have origins and characteristics that are different from those that occur under present-day conditions."

Lines 286-288. This sentence is correct in detail (Bozkurt et al. did evaluate the impact on rainfall had the SST been lower in late March 2015), but contorts the logic relative to the purpose to Reyers et al. The reader must know independently that the SST was anomalously high during late March 2015 in order to understand that Bozkurt et al. study a real versus hypothetical system opposite to the current study. I think this can be rewritten for greater clarity.

Yes, we add details: „(…) Bozkurt et al. (2016) show that a hypothetical reduction of the SST in the eastern tropical Pacific during the March 2015 rainfall event, which occurred during anomalously high SST, significantly decreases the precipitable water (…)"

Line 286. Bozkurt et al. specified that there was uncertainty about how SST impacted the March 2015 event, whether through activity in the marine boundary layer or through convection. This new paper specifies that the MCB is not expressed in the marine boundary layer. Is there a value to the authors reflecting back on this aspect of the Bozkurt et al. paper?

We add: „ Our results support that higher SSTs lead to stronger rainfall in the Atacama, broadly consistent with the March 2015 case studied by Bozkurt et al. (2016). "

Line 295. A verb and adverb are missing. Insert "are" in the phrase "occur in the mid-Pliocene but ARE not PRESENT during present-day conditions".

Thanks, we add verb and adverb.

Lines 300-304. It should be noted, however, that the isotopic composition of the March 2015 rain at coastal sites clearly indicates that the rain in that event had a tropical Pacific water source. This has not been made compatible with the B2021 atmospheric models. (Data presented in Jordan et al., 2019).

We add: „ Interestingly, the isotopic analysis of the collected rain water in coastal regions from the March 2015 event indicates a tropical Pacific origin for the water (Jordan et al., 2019). While B2021 did not specifically address this particular event, they show tropical and subtropical Pacific origins for cases that are weaker in terms of humidity along the identified trajectories. The uncertainty to identify the most representative target location and time for the trajectory calculation may cause stronger events to appear weaker. "
Further research, e.g., using more coinciding isotopic and trajectory analyses, would be helpful to clarify ultimately.

„Conclusions and Outlook
Line 308. As a generality, in the Conclusions the authors should speak of the full names of features rather than using acronyms. Some readers will only look at the Conclusions, not all the preliminary material, and they will not gather anything useful by learning that MCBs did something. They will learn more if the statement is that Moisture Converyor Belts did something."

Agreed, we introduce MCB, SST, and ENSO in the revised conclusion. We also add more information on the implication of the work in the context of newly available literature: „ Our regional evaluation is interesting in the context of the relatively high climate sensitivity of CESM2, which might be seen as an outlier in a larger ensemble of CMIP6 simulations for other time periods (Burls and Sagoo, 2022). It was proposed to use paleo-simulations as testbed for climate model performance to constrain climate sensitivity (Burls and Sagoo, 2022, Zhu et al., 2022). Our results suggest that paleo-simulations paired with regional downscaling to kilometre-scales might also be useful for better understanding and predicting regional climate changes with global warming, e.g., concerning the hydrological cycle that remains an outstanding challenge for global models with parameterised convection. If our mid-Pliocene simulation is a useful out-of-sample test, the fact that CESM2 outperforms other models with lower climate sensitivity for the mid-Pliocene climate in the region of the Atacama Desert would support a high climate sensitivity."

„Lines 391-395. The Bozkurt et al. references is duplicated."

Thanks, we remove the duplication.

---

## Author Comment (AC3)

**Point-by-point reply for Reyers et al.:**
**„On the importance of moisture conveyor belts from the tropical East Pacific for wetter conditions in the Atacama Desert during the Mid-Pliocene"**

We thank all community members who provided comments on our manuscript for the appraisal of our manuscript. The comments helped us to further improve the presentation of the results in the manuscript. Our replies to the comments along with details on how we intend to revise the manuscript are printed in blue below the original comments in black. We also revise the color schemes in the figures for clarity.

**Reply to RC2 by an anonymous referee**

„I found this work very novel and interesting to read. The focus on moisture conveyer belts in the Atacama during the mid-Pliocene is an important contribution to understand the mechanisms of past and present rainfall events in the region. The experimental design is well accomplished, and I liked very much the use of SOM and clustering techniques for MCBs detection and pattern analysis."

Thank you for your appraisal of the manuscript.

„I have some general comments for different sections of the manuscript:
Introduction Authors mention that the increased rainfall in the Southern Atacama Desert is mostly duo to a northward displacement of mid-latitudinal westerlies and extra-tropical winter cyclones. In my opinion they cite literature that does not support this statement. For example, they cite Jordan et al., 2019 as evidence of southwestern moisture source but Jordan et al., 2019 identifies the tropical Pacific as the main moisture source of the March 2015 extreme rainfall event. Can please the authors clarify this inconsistency. Also, I noticed that Bartz et al., 2019 do not actually state a southwestern moisture source in their study, the same with Stuut and Lamy, 2017."

The thought of a southwestern moisture source was based on the following statements in the papers:

- **Bartz et al. 2019** mention: *„Thus, based on our observations and in comparison with marine palaeoclimate records "…", alluvial fan dynamics along the western flank of the Coastal Cordillera seems to be influenced by an interplay between northward-driven austral Westerlies, ENSO related positive SST anomalies, and variations in the strength and the position of the SE Pacific anticyclone.", which suggests a southwestern moisture source.*
- **Stuut and Lamy, 2017**: *"A tendency toward more El Niño-like conditions would be consistent with more humid conditions in northern Chile, as at present, within the northern winter rain belt of Chile, strong positive rainfall anomalies occur during El Niño events induced by a northward shift of the Southern Westerlies due to a weakening and northward displacement of the SE Pacificanticyclone (Ruttland and Fuenzalida, 1991).", which also suggests a southwestern moisture source.*
- **Jordan et al., 2019**: *"South of 22° S (northern part of the political division "II Region" of Antofagasta"), Pacific-sourced water vapor leads to precipitation in the Andes Mountains dominantly in winter (June-July-August) (zone III) (Houston and Hartley, 2003; Burgener et al., 2016). Through cutoffs and fronts from the mid-latitude westerlies (Vuille and Ammann, 1997) a decreasing amount of precipitation reaches progressively northward."* We remove the citation of Jordan et al. (2019) and add Vuille and Ammann, 1997.

The revised manuscript text is: "Intervals of increased rainfall in the Southern Atacama Desert are mostly attributed to a northward displacement of mid-latitudinal westerlies and accompanied extra-tropical winter cyclones (Vuille and Ammann, 1997, Stuut and Lamy, 2017; Bartz et al., 2019), which suggest a southwestern moisture source."

In line #80 authors state the hypothesis of the tropical Southeast Pacific as a moisture source for the Atacama but this was demonstrated in Bozkurt et al., 2016. It is possible to clarify how their hypothesis differs from the mechanism that triggered the events of March 2015? In its present writing form, it is not obvious the connection with Bozkurt et al., 2016's findings.

Indeed, the mechanisms identified by Bozkurt et al. (2016) for the March 2015 severe rainfall event in the Atacama could be an important mechanism in the past climate. However, the past and present constellations of the global atmospheric and oceanic circulations are substantially different and it remains to be tested whether the processes responsible for the March 2015 rainfall event are also statistically significant for a wetter Atacama in the Mid-Pliocene. Our high-resolution simulations for the Mid-Pliocene indicate that the essence of these mechanisms may also be importance in the paleoclimate context.

We modify the text to reflect this point: "The tropical Southeast Pacific northwest of the desert could be a potential moisture source for increased humidity in the mid-Pliocene, like assessments of the regional rainfall under present-day climate suggest (Bozkurt et al., 2016, Jordan et al., 2019; Böhm et al., 2021). However, the past and present constellations of the global atmospheric and oceanic circulations are substantially different."

We also add in the conclusion: "Our results support that higher SSTs lead to stronger rainfall in the Atacama, broadly consistent with the March 2015 case studied by Bozkurt et al. (2016)."

„Data and Methods
Can the authors please explain why using orbital parameters from the pre-industrial period and not the orbital parameters of the mid-Pliocene. Orbital forcing of later periods has proved to be useful in reproducing past climates. For example, Engelbrecht, F. A., and Coauthors, 2019: Downscaling Last Glacial Maximum climate over southern Africa. Quat. Sci. Rev., 226, https://doi.org/10.1016/j.quascirev.2019.105879. I understand that PlioMIP simulations use orbital parameters for 1850 but it would be very useful for the non-specialized community to understand why we are modelling the climate of mid-Pliocene using orbital parameters for present day. This forcing is not negligible as discussed by Willet et al., 2013 (Willeit, M., A. Ganopolski, and G. Feulner, 2013: On the effect of orbital forcing on mid-Pliocene climate, vegetation. Clim. Past, 9, 1749–1759, https://doi.org/10.5194/cp-9-1749-2013). This is important for ice sheets extension and therefore albedo and the global energy balance."

We chose the setup of the regional climate model to be as close as possible to the global PlioMIP2/PMIP4 experiment to ensure consistency across the model chain. In our regional experiment for the Atacama region, we have no large ice sheets that could be affected by this choice, although we agree that this aspect should be revisited when new global climate simulation for the Pliocene will be conducted in the future. We have added: „ (…) orbital parameters are as for the pre-industrial period (1850) to be consistent with the setup of PlioMIP2 experiments"

„What is the actual bias of WRF historical run? As precipitation is very reduced in the hyper-arid core of the Atacama, simulated vs observed precipitation can have many orders of magnitude of difference. This is not a problem and is common in modelling studies, but I missed a more robust measure of uncertainty of modelling experiments using WRFhist. "

We evaluated the rainfall from $WRF_{hist}$ against a WRF simulation that downscales the ERA5 reanalysis for the same domain and spatial resolution ($WRF_{era}$). The results for the annual and seasonal mean precipitation patterns along with limitations are shown in Fig. 3 and are mentioned in Section 3.1. We now revise the paragraph to better highlight the evaluation results: "We evaluated the rainfall from $WRF_{hist}$ against a WRF simulation that downscales the ERA5 reanalysis for the same domain and spatial resolution ($WRF_{era}$). There are quantitative differences in rainfall,

but the aridity is overall satisfyingly reproduced by the WRF simulation that used data from the historical simulation of CESM2 at the lateral boundaries (WRF$_{hist}$). Specifically, the spatial patterns and the seasonal cycle of rainfall are qualitatively captured by WRF$_{hist}$ (compare Fig. 3f-j with Fig. 3a-e). Both WRF$_{era}$ and WRF$_{hist}$, show (…) Annual and seasonal rainfall amounts tend to be regionally overestimated by WRF$_{hist}$ against WRF$_{era}$, but the hyper-aridity with only a few mm of rainfall per year is well simulated (Fig. 3f-j). We therefore conclude that the WRF simulations using CESM2 as boundary conditions are suitable for our research interest."

„Results
It is not clear to me which proxy data was used to validate model projections. Maybe these is all due to the lack of proxy records for such a long period of time. I think this is important since the authors assure that CESM2 agrees with reconstructions, but they don´t provide any evidence of to which extent the model agrees with proxy data. The only reconstructions available are those provided by Dowsett et al., 2013?"

There are more proxy data available. We add the new table below to summarize geological records from the wider study area that fall into the mid-Pliocene. The table contains details on the interpreted proxy data and statements on the wetter conditions relative to present-day, broadly consistent with CESM2 that we use as boundary data for our regional kilometer-scale simulation.

| Name of site | Coordinates | Time period | Type of proxy data | Signal relative to modern climate | Reference |
|---|---|---|---|---|---|
| Cerro Soledad, Quillagua-Llamara basin | 21.25° S; 69.5° W | 3.2–2.7 Ma | CN dating of lake terraces | Wetter conditions in the Altiplano | Ritter et al. (2018) |
| Soledad Fm, Quillagua-Llamara basin | 20-21° S; 69-70° W | 4.2-2.6 Ma | ash layers in playa-lake sediments | Wetter conditions in the Altiplano | Vásquez et al. (2018) |
| Tiliviche Paleolake | 19.5° S; 70° W | 3.5-~3.0 Ma | salar deposits in the Tivliche paleolake | Wetter conditions in the Altiplano | Kirk-Lawlor et al. (2013) |
| Lauca basin | 18.5° S 69.25° W | 3.7–2.6 Ma | lacustrine and fluvial sediments | Local proxy for semi-arid conditions with increased precipitation | Gaupp et al. (1999) |
| Cordillera de la Sal, Salar de Atacama basin | 23° S 68.25° W | 3.5 – 2 Ma | lacustrine and mudflat deposits | Wetter conditions in the Cordillera | Evenstar et al. (2016) |
| Calama Basin | 22.5° S 69° W | 6 – 3 Ma | palustrine carbonates | Wetter conditions in the Altiplano | May et al. (2005) |
| Central Depression, Calama basin, and Preandean Depression | 19.75 −23° S | 8 – 3 Ma | fluviolacustrine and alluvial-fan deposits | Semi-arid conditions | Hartley & Chong (2002) |
| Coastal Cordillera draianges | 23.45 - 29.9° S | > 2.1 Ma | CN dating and near surface ash ages | Wetter conditions | Amundson et al. (2012) |

**Table 1: Proxy data for wetter condition than present-day in the region of the Atacama Desert that fall into the mid-Pliocene.**

The new table is referenced in the results: "These results for more rainfall are broadly consistent with proxy records for the wetter conditions in the mid-Pliocene compared to pre-industrial in the region, listed in Table 1. "

We further add citations for proxy data on the SST difference between the mid-Pliocence and present-day: "The model results are supported by proxy data indicating a global SST anomaly for the mid-Pliocene vs. pre-industrial of 2.3°C and 3.2–3.4°C based on foraminifera Mg/Ca and alkenones or alkenones only, respectively (McClymont et al., 2020). Specifically in the upwelling regions at the Peruvian margin, Deckens et al. (2007) reconstructed a Pliocene-modern SST change by 2.9°C"

„Still, if possible, authors can provide a measure of uncertainty in their modelling design. In modelling experiments for future projections, as an example, is very important to measure the level of uncertainty and therefore the model ensemble is used, and a range of possible climates is provided. I can guess authors did not use the ensemble because the mean precipitation tended to be lower than current climate (?). Still, the question is, if only one model is used, how can we be sure that CESM2 model results are not due to chance? At least authors should mention the limitations of using only 1 model."

We decided to perform a regional downscaling experiment from global model output that showed the expected difference in the mean state between the mid-Pliocene and present-day. It would indeed be great to have more PlioMIP2 model simulations with the expected changes to assess to what extend our results are influenced by model-to-model differences. More paleo-simulations would be useful as testbed for model simulations for modern climate change, but running more models for paleo-climate seems difficult, especially for those models that have a high climate sensitivity like CESM2 (Burls and Sagoo, 2022). It would be valuable to have data from more global model simulation for the Pliocene or other warm climates available in the future. We talk about this aspect now in the conclusion: „ Our regional evaluation is interesting in the context of the relatively high climate sensitivity of CESM2 (Gettelman et al., 2019), which might be seen as an outlier in a larger ensemble of CMIP6 simulations for other time periods (Burls and Sagoo, 2022). It was proposed to use paleo-simulations as testbed for climate model performance to constrain climate sensitivity (Burls and Sagoo, 2022, Zhu et al., 2022). Our results suggest that paleo-simulations paired with regional downscaling to kilometre-scales might also be useful for better understanding and predicting regional climate changes with global warming, e.g., for the hydrological cycle that remains an outstanding challenge for global models with parameterised convection. If our mid-Pliocene simulation is a useful out-of-sample test, the fact that CESM2 outperforms other models with lower climate sensitivity for the mid-Pliocene climate in the region of the Atacama Desert would support a high climate sensitivity. It would be valuable to have data from more global model simulation for the Pliocene or other warm climates for similar downscaling experiments in future research, especially from CMIP6 models with a high climate sensitivity. This endeavour requires also further development of proxy data for paleo climates, of which there are still a limited number for the Pliocene."